# ODE-GS: Latent ODEs for Dynamic Scene Extrapolation with 3D Gaussian Splatting

**Daniel Wang**[1]   **Patrick Rim**[1]   **Tian Tian**[2]   **Dong Lao**[3]   **Alex Wong**[1]   **Ganesh Sundaramoorthi**[4]

[1]Yale University   [2]TU Delft   [3] Louisiana State University   [4]RTX

[1]{daniel.wang.dhw33, patrick.rim, alex.wong}@yale.edu

[2]T.Tian@student.tudelft.nl  [3]dlao1@lsu.edu   [4]ganesh.sundaramoorthi@rtx.com

## Abstract

We introduce ODE-GS, a novel approach that integrates 3D Gaussian Splatting with latent neural ordinary differential equations (ODEs) to enable future extrapolation of dynamic 3D scenes. Unlike existing dynamic scene reconstruction methods, which rely on time-conditioned deformation networks and are limited to interpolation within a fixed time window, ODE-GS eliminates timestamp dependency by modeling Gaussian parameter trajectories as continuous-time latent dynamics. Our approach first learns an interpolation model to generate accurate Gaussian trajectories within the observed window, then trains a Transformer encoder to aggregate past trajectories into a latent state evolved via a neural ODE. Finally, numerical integration produces smooth, physically plausible future Gaussian trajectories, enabling rendering at arbitrary future timestamps. On the D-NeRF, NVFi, and HyperNeRF benchmarks, ODE-GS achieves state-of-the-art extrapolation performance, improving metrics by $19.8\%$ compared to leading baselines, demonstrating its ability to accurately represent and predict 3D scene dynamics.[1]

## 1 Introduction

Recently, 3D Gaussian Splatting (3DGS) methods have emerged as an effective approach for dynamic scene reconstruction. By training on images taken from a time-dependent 3-dimensional (3D) scene, such methods enable photorealistic novel view synthesis (NVS) for any time within the observed window. However, the task of *prediction*—extrapolating future scene dynamics from past observations—remains largely underexplored. Performing such a task is a well-studied capability that humans possess (Rao & Ballard, 1999; Mrotek & Soechting, 2007; Battaglia et al., 2013; Khoei et al., 2017), while mirroring this capability in intelligent systems is of great interest for applications such as self-driving, robotics, and augmented reality. Our focus is then to bridge this gap and enable the ability to forecast future 3D states in the context of dynamic scene reconstruction, which we will refer to as **dynamic scene extrapolation**.

Performing dynamic scene extrapolation is fundamentally more challenging than reconstruction. Existing methods are primarily designed for temporal *interpolation* within the observed window, where arbitrary view points at given time can be reconstructed by conditioning models on timestamps. In contrast, temporal *extrapolation* is inherently under-constrained, as there exist infinitely many possible future dynamics given our past observations. Therefore, popular dynamic scene reconstruction methods, such as TiNeuVox (Fang et al., 2022), Deformable 3D Gaussians (Yang et al., 2024), and 4D Gaussian Splatting (Wu et al., 2024), excel at "filling the gaps" between observed timestamps, but degrade when extended beyond them. In such cases, future timestamps fall outside the training distribution, leading to out-of-distribution (OOD) failures.

As with many under-constrained problems, our task then becomes estimating the *most likely* future dynamics given existing observations. Incorporating physical assumptions can substantially narrow this solution space. An example of such an assumption is the spatio-temporal smoothness of motion, which provides strong constraints on the evolution of scene dynamics. Differential equations have long served as principled tools for describing the evolution of physical systems (Chen et al., 2018),

---

[1]Code available at: https://github.com/preacherwhite/ODE-GS

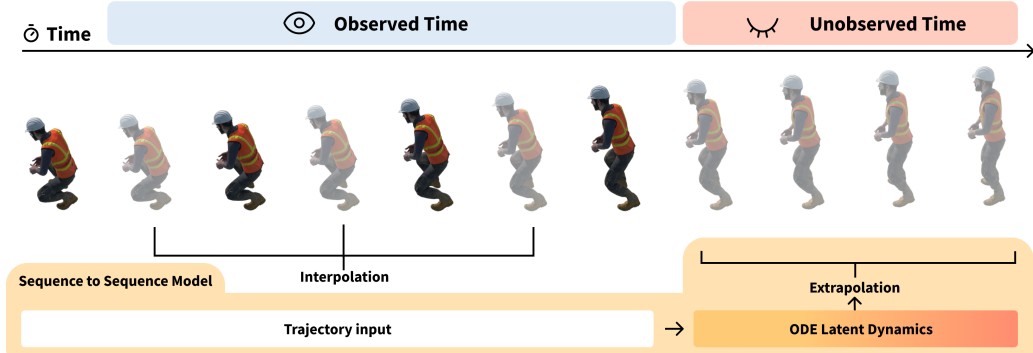

Figure 1: Unlike existing methods that focus on interpolation, i.e., reconstructing novel scene views at unseen timestamps *within the observed time window*, we focus on extrapolation, i.e., extending scene dynamics beyond the observed times, by first training a representation of the observed scene and then using a sequence-to-sequence model to reconstruct *future* novel views via latent ODE dynamics.

and **ordinary differential equations (ODEs)** in particular offer a natural formalism for representing continuous and physically plausible motion trajectories.

In light of these considerations, we propose to represent the temporal evolution of a dynamic scene in a continuous-time latent space, where the evolution is governed by an ODE. In this way, we impose a physical assumption of smooth motion into the predictive model (Chen et al., 2018; Rubanova et al., 2019), while retaining 3DGS's capability of high-fidelity rendering. Unlike prior approaches that condition directly on explicit timestamps—leading to OOD failures—we reformulate dynamic scene extrapolation as a sequence-to-sequence forecasting problem, which naturally aligns with 3DGS's explicit representation of the 3D scene. Specifically, we first encode a temporal sequence of Gaussian parameters using a Transformer (Vaswani et al., 2017) into a latent state that embeds past motion. We then model the temporal evolution of this latent state via a neural ODE, where a neural network parameterizes its velocity field. Extrapolation is thus realized by numerically integrating the latent dynamics beyond the observed window, yielding future latent states. These states are subsequently decoded back into Gaussian parameters, which can be rendered into novel future views. To further constrain extrapolation, we incorporate additional physical priors as lightweight regularizers during training. These include penalties that encourage smoothness in both the latent trajectories and the decoded Gaussian parameters, reinforcing the assumption of continuous motion.

The resulting method, **ODE-GS** (**O**rdinary **D**ifferential **E**quation-based **G**aussian **S**platting), decouples scene reconstruction from temporal forecasting. We first optimize a Gaussian interpolation model within the observed window, by training a set of canonical Gaussian parameters and a deformation network. This interpolation model is then frozen and used as a data generator, producing temporal trajectories of 3D Gaussian parameters. We train the Transformer-Latent ODE on these trajectories, but crucially, the model is conditioned only on a partial prefix of each sequence and learns to extrapolate the remainder. This training setup equips ODE-GS to extrapolate beyond the observed time horizon at inference, by integrating forward in time via our latent ODE and decoding back into future Gaussian parameters.

Our contributions are:

- We propose **ODE-GS**, which integrates 3D Gaussian Splatting with a Transformer-based latent ODE. We model scene dynamics as continuous trajectories in latent space to enforce smoothness priors, enabling stable extrapolation beyond the observed window.

- We validate our modeling strategy that decouples dynamic scene reconstruction from temporal forecasting. By first optimizing an interpolation model for reconstruction and then training the latent ODE forecaster on generated trajectories, ODE-GS avoids direct reliance on timestamp conditioning and mitigates out-of-distribution failures.

- We incorporate inductive biases as regularizers that encourage smoothness in both latent and decoded trajectories, improving stability and rendering quality at extrapolated timestamps.

- ODE-GS achieves state-of-the-art performance in dynamic scene extrapolation, improving over the best existing extrapolation method by an average of 21.4% PSNR, 7.4% SSIM, and 30.5% LPIPS across synthetic (D-NeRF, NVFi) and real-world (HyperNeRF) datasets.

## 2 RELATED WORK

**Novel View Synthesis (NVS).** Recent advances in NVS have explored a diverse range of approaches, including explicit mesh-based representations (Broxton et al., 2020; Dou et al., 2016; Newcombe et al., 2015; Orts-Escolano et al., 2016), and implicit neural volume representations (Lombardi et al., 2019). Among these, Neural Radiance Fields (NeRF) (Mildenhall et al., 2021; Duan & Wong, 2026) have emerged as a dominant paradigm due to their leading performance in NVS. The foundational success of NeRF has led to numerous extensions for dynamic scene reconstruction (Attal et al., 2023; Du et al., 2021; Ost et al., 2021; Park et al., 2021a; Pumarola et al., 2021), enabling applications such as monocular video-based scene reconstruction (Gao et al., 2021; Li et al., 2021; Tretschk et al., 2021), editable scene representations (Kania et al., 2022; Park et al., 2021a), and human-centered reconstructions (Peng et al., 2021b;a). Notably, NVFi (Li et al., 2023) investigates the problem of dynamic scene extrapolation by adding geometric priors. However, its dependence on explicit timestamps during training induces out-of-distribution errors when extrapolating.

**3DGS and Dynamic Scene Modeling.** 3DGS (Kerbl et al., 2023), first used for static NVS, has become a popular choice for representing dynamic scenes due to its speed and explicit nature (Huang et al., 2024; Li et al., 2024; Duan et al., 2024). Deformable 3D Gaussians (Yang et al., 2024) learn Gaussians in a canonical space with a deformation network. By training both the canonical Gaussians and the deformation network simultaneously, they enable continuous-time rendering by transforming canonical Gaussians into arbitrary time within the training window. Concurrently, 4D Gaussian Splatting (Wu et al., 2024) took the same canonical-deform strategy and additionally integrated 4D neural voxels inspired by HexPlane (Cao & Johnson, 2023). Additional efforts, such as explicit time-variant Gaussian features (Luiten et al., 2024), achieved interactive frame rates and enabled flexible editing. GaussianVideo (Bond et al., 2025) uses neural ODEs, but for learning smooth camera trajectories rather than scene motion. However, most 3DGS-based approaches rely on a time-conditioned deformation field, making these models excel in interpolation tasks but unable to extrapolate into unseen time in the future. GaussianPrediction (Zhao et al., 2024) has recently explored this issue by combining a superpoint strategy with Graph Convolution Networks (GCN) that directly conditions on past motion instead of time, but is only capable of sampling at discrete steps when extrapolating.

**Neural Ordinary Differential Equations.** Neural Ordinary Differential Equations (Neural ODEs) (Chen et al., 2018) introduced a novel approach for continuous-depth neural networks. Instead of defining discrete layers, a Neural ODE specifies the continuous dynamics of a hidden state using a neural network that parameterizes its derivative. The network's output is then determined by a numerical ODE solver that integrates these learned dynamics over a specified interval. Key advantages include memory-efficient training via the adjoint sensitivity method, inherent handling of irregularly-sampled data, and adaptive computation.

Relevant to dynamic modeling, Latent ODEs (Rubanova et al., 2019) typically encode an input sequence into an initial latent representation whose continuous-time evolution is then governed by a Neural ODE. A decoder can subsequently map these evolving latent states back to the observation space at arbitrary times. This is effective for modeling continuous trajectories and is often combined with Variational Autoencoders (VAEs), as in ODE2VAE (Yildiz et al., 2019), to learn distributions over latent paths and capture uncertainty.

The synergy between recurrent methods and Neural ODEs has also been explored. For instance, GRU-ODE (De Brouwer et al., 2019) adapts GRU-like gating mechanisms to continuously evolving states, while ODE-RNN (Rubanova et al., 2019) interleaves discrete RNN updates at observation points with continuous ODE-based evolution. Recent methods have explored using Neural ODEs for vision tasks like images (Liu et al., 2025), but applications in 3D representation are still limited.

## 3 METHODOLOGY

In the following sections, we formalize our problem setting and detail each component of ODE-GS: first, the interpolation model that generates dense Gaussian trajectories within the observed window (Sec. 3.1); second, the Transformer-based latent ODE architecture for extrapolation (Sec. 3.2); third, a dynamic sampling strategy to train the extrapolation model on different forecasting horizons (Sec. 3.3); and finally, the training and regularization objectives that ensure physically plausible and stable extrapolated dynamics (Sec. 3.4). Implementation details are discussed in Sec. B.1.

**Notation and Formalization**. Let $\{I_i\}$ be a set of calibrated RGB images, with corresponding camera poses $\{V_i\}$ capturing a dynamic 3D scene at timestamps $\{t_i\}$,

$$\mathcal{D} = \{(I_i, V_i, t_i)\}_{i=1}^N, \quad I_i : \mathbb{R}^{3 \times H \times W}, \quad V_i \in SE(3), \quad t_i \in \mathbb{R}, \tag{1}$$

we aim to learn a continuous-time rendering operator $\mathcal{F} : \mathbb{R} \times SE(3) \to \mathbb{R}^{3 \times H \times W}$ that generates RGB images for any time $t$ and camera pose $V$. The operator decomposes as:

$$\mathcal{F}(t, V) = \mathcal{R}(\mathcal{G}(t), V), \quad \mathcal{G}(t) = \begin{cases} \overline{\mathcal{G}} + \mathcal{D}_\omega(t, \overline{\mathcal{G}}) & \text{if } t_{\min} \leq t \leq t_{\max}, \\ \mathcal{E}_\phi(\gamma, t) & \text{if } t > t_{\max}, \end{cases} \tag{2}$$

where $\overline{\mathcal{G}} = \{\overline{G}_k\}_{k=1}^M$ is a learnable canonical set of 3D Gaussians, $\mathcal{D}_\omega$ is an interpolation deformation function, $\mathcal{E}_\phi$ is a sequence-to-sequence extrapolation model with Transformer Latent ODE architecture, and $\mathcal{R}$ is the differentiable rasterizer from (Kerbl et al., 2023). $\gamma$ is a sequence of past Gaussian parameters that serve as input, which is detailed in Sec. B.

Each canonical Gaussian $\overline{G}_k = (\mu_k, q_k, s_k, c_k, \alpha_k)$ comprises position $\mu_k \in \mathbb{R}^3$, quaternion $q_k \in \mathbb{R}^4$, scales $s_k \in \mathbb{R}^3$, opacity $\alpha_k \in \mathbb{R}$, and spherical harmonics (SH) coefficients $c_k \in \mathbb{R}^d$, where $d$ is the number of SH functions used, usually set to 3. For each $k$, we keep $\alpha_k$ and $c_k$ consistent across time, so that only $\mu_k, q_k$, and $s_k$ are time-dependent. For simplicity, from now on, $G(t)$ refers to only these three parameters. We may then derive the rotation matrix $R_k \in \mathbb{R}^{3 \times 3}$ from $q_k$, the scaling diagonal matrix $S_k \in \mathbb{R}^{3 \times 3}$ from $s_k$, and the covariance matrix for each Gaussian by $\Sigma_k = R_k S_k S_k^\top R_k^\top$.

The differentiable rasterizer (Kerbl et al., 2023) $\mathcal{R}$ renders images by projecting each 3D Gaussian onto the image plane, computing per-pixel alpha compositing with front-to-back blending:

$$C(p) = \sum_{k \in G(p)} c_k \alpha_k \prod_{j=1}^{k-1} (1 - \alpha_j), \tag{3}$$

where $G(p)$ are Gaussians affecting pixel $p$, $c_k$ is the color, and $\alpha_k$ is the opacity after 2D projection. We first learn the interpolation model $\overline{\mathcal{G}}$ and $\mathcal{D}_\omega$, then freeze them and train $\mathcal{E}_\phi$ for extrapolation.

### 3.1 INTERPOLATION MODEL FOR TRAJECTORY GENERATION

To obtain trajectories of Gaussian parameters within the observed window, we adopt a canonical-plus-deformation strategy (Yang et al., 2024; Wu et al., 2024) that has become standard in dynamic scene modeling. A canonical set of 3D Gaussians, $\overline{\mathcal{G}}$, represents the static reference configuration of the scene, while a lightweight, time-conditioned deformation Multi-Layer-Perceptron (MLP) $\mathcal{D}_\omega$ predicts offsets for position, rotation, and scale at each timestamp $t$. This enables continuous interpolation of Gaussian states across time. The interpolation model is trained using a photometric reconstruction objective:

$$\mathcal{L}_{\text{render}} = (1 - \lambda) \cdot \|\widehat{I}_i - I_i\|_1 + \lambda \cdot (1 - \text{SSIM}(\widehat{I}_i, I_i)), \tag{4}$$

Where $\widehat{I}_i$ is rendered using the differentiable rasterizer: $\widehat{I}_i = \mathcal{R}(\overline{\mathcal{G}} + \mathcal{D}_\omega(t, \overline{\mathcal{G}}), V_i)$. After training, we freeze both the canonical Gaussians $\overline{\mathcal{G}}$ and the deformation network $\mathcal{D}_\omega$, so that given any time $t$ within the observed time window, $G_k(t)$ may be generated for all $k$. SSIM refers to the Structural Similarity Index Measure.

### 3.2 LATENT ODE MODEL FOR EXTRAPOLATION

Viewing temporal prediction as a mapping between an observed sequence and a future sequence, we model scene dynamics as a sequence-to-sequence problem. Our extrapolation module $\mathcal{E}_\phi$ is a Transformer Latent ODE that predicts future dynamics from past Gaussian trajectories.

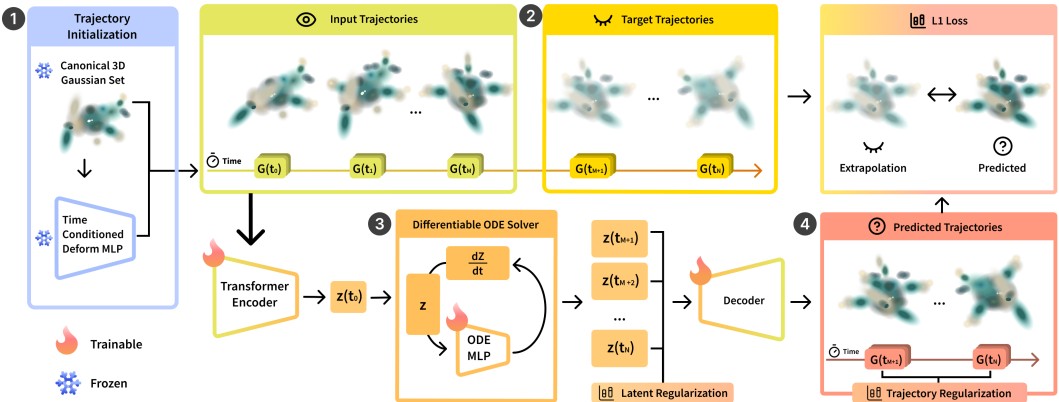

Figure 2: **1:** We initialize temporal trajectories of 3D Gaussian parameters using the frozen interpolation model, which consists of the canonical 3D Gaussian set and a time-conditioned deformation MLP. These trajectories lie entirely within the observed temporal window. **2:** Through our dynamic sampling strategy, each Gaussian trajectory is sampled into multiple observed prefix (input) and a held-out suffix (target) trajectories, providing training pairs for the Transformer latent ODE. **3:** Latent-ODE training encodes the observed prefix with a Transformer, infers a latent initial state, and evolves it forward with a neural ODE. **4:** A decoder maps the latent path back to Gaussian parameters, which are supervised against the ground-truth suffixes via an L1 loss and smoothness regularizers.

Given an input sequence

$$\gamma_k = \{G_k(t_j)\}_{j=1}^{N_c},$$

uniformly sampled from a context window of length $N_c$ for Gaussian $k$, we embed each step and add sinusoidal positional encodings to preserve temporal order. The resulting sequence is processed by a Transformer encoder

$$\mathcal{F}_\phi : \mathbb{R}^{N_c \times 10} \to \mathbb{R}^d,$$

yielding a latent representation $z(t_0) \in \mathbb{R}^d$ that summarizes past dynamics. This latent state initializes a neural ODE, parameterized by an MLP:

$$\dot{z} = \frac{dz}{dt} = f_\theta(z(t)).$$

Numerical integration produces a continuous latent trajectory $z(t)$ for any $t > t_{\max}$. A decoder then maps the evolved latent states back to Gaussian parameters:

$$\delta_\psi : \mathbb{R}^d \to \mathbb{R}^{10}, \hat{G}_k(t) = \delta_\psi(z(t)).$$

This combination of sequence encoding and continuous latent evolution allows $\mathcal{E}_\phi$ to generate smooth Gaussian trajectories without explicit timestamp embeddings, enabling extrapolation to arbitrary future horizons where unseen timestamps are no longer out-of-distribution.

### 3.3 DYNAMIC TRAJECTORY SAMPLING

To effectively train the extrapolation module, it is essential to expose the model to prediction tasks spanning a wide range of forecasting horizons. Therefore, unlike common approaches where sampled trajectories always occupy the same time span uniformly (Li et al., 2023), we design a dynamic trajectory sampling strategy. The pre-trained interpolation model provides continuous trajectories of Gaussian parameters, from which we extract an observed prefix and a future suffix. The prefix is sampled at fixed intervals to ensure consistent input dimensionality, while the suffix varies in temporal span depending on the selected starting time.

Our training dataset is constructed as the union over all possible prefix–suffix splits of Gaussian trajectories across all Gaussians and starting times. This design compels the extrapolation model to learn from both short-term and long-term forecasting instances within a unified training procedure, thereby encouraging robust generalization beyond the observed window, as described in Sec. B. At

inference, the model conditions on the final observed prefix and extrapolates Gaussian trajectories forward in time, which are subsequently rendered into novel frames. At test time, we take the *final context segment* of length $N_c$ from the observed window, generated by the interpolation model $\mathcal{D}_\omega$. This sequence is encoded and evolved forward by the Latent ODE to produce extrapolated Gaussians $\hat{G}_k(t)$ for $t > t_{\max}$. The full scene $\hat{\mathcal{G}}(t) = \{\hat{G}_k(t)\}_{k=1}^M$ is rendered with a differentiable rasterizer:

$$\widehat{I}(t, V) = \mathcal{R}(\hat{\mathcal{G}}(t), V).$$

### 3.4 TRAINING OBJECTIVE

**Extrapolation Loss.** Given a training pair consisting of a context trajectory $\gamma_c$ and its corresponding target trajectory $\gamma_e = \{G_k(t_j)\}_{j=1}^{N_e}$, the extrapolation module $\mathcal{E}_\phi$ produces predictions $\hat{\gamma}_e = \{\hat{G}_k(t_j)\}_{j=1}^{N_e}$. The extrapolation loss supervises these predictions by minimizing the mean absolute error (L1) between target Gaussian parameters generated by the interpolation model and the predicted parameters by the extrapolation model across the target temporal window:

$$\mathcal{L}_{\mathrm{e}} = \frac{1}{N_e} \sum_{j=1}^{N_e} \left\| \hat{G}_k(t_j) - G_k(t_j) \right\|_1. \tag{5}$$

This objective ensures that the predicted Gaussian trajectories align closely with true future dynamics before additional regularization terms are applied.

To ensure physical plausibility and prevent overfitting to the observed trajectory, we also introduce two complementary regularizations that promote smoothness in latent dynamics and 3D trajectories.

**The latent regularization** penalizes high-frequency oscillations in the learned ODE function. Given the latent trajectory $z(t)$ evolved by the neural ODE with velocity field $\dot{z}(t) = f_\theta(z(t))$, we approximate the latent acceleration through finite differencing:

$$\mathcal{R}_{\mathrm{latent}} = \frac{1}{N_e - 1} \sum_{j=1}^{N_e - 1} \left\| \frac{f_\theta(z(t_{j+1})) - f_\theta(z(t_j))}{\Delta t_j} \right\|_2^2 \tag{6}$$

where $\{t_j\}_{j=1}^{N_e}$ are the extrapolation timestamps, $\Delta t_j = t_{j+1} - t_j$ the step size, $f_\theta$ the neural ODE.

**The trajectory regularization** enforces smoothness directly in 3D space by penalizing accelerations of the Gaussian positions. For each Gaussian $G_k(t)$, $\mu_k(t) \in \mathbb{R}^3$ is its position at time $t$. We compute:

$$\mathcal{R}_{\mathrm{traj}} = \frac{1}{N_e - 2} \sum_{j=1}^{N_e - 2} \left\| \frac{v_k(t_{j+1}) - v_k(t_j)}{\Delta t_j} \right\|_2^2 \tag{7}$$

the per-Gaussian velocity $v_k$ is approximated as:

$$v_k(t_j) = \frac{\mu_k(t_{j+1}) - \mu_k(t_j)}{\Delta t_j} \tag{8}$$

**Adaptive weighting for regularization.** In early stages, strong regularization may inhibit the model from learning meaningful dynamics. To address this, we introduce an *adaptive regularization weighting mechanism* that dynamically adjusts the contribution of regularization throughout training. At each iteration, we estimate the model's convergence state using the Exponential Moving Average (EMA) of the trajectory prediction loss, which provides a stable signal compared to the raw loss that may fluctuate due to oscillations. As training progresses and the extrapolation loss decreases, the regularization weight is increased. This gradually biases the model toward selecting smoother trajectories among the many plausible solutions, thereby guiding fine-grained trajectory predictions toward stable convergence. The final training loss then becomes:

$$\mathcal{L} = \mathcal{L}_{\mathrm{e}} + s_t (\lambda_{\mathrm{latent}} \mathcal{R}_{\mathrm{latent}} + \lambda_{\mathrm{traj}} \mathcal{R}_{\mathrm{traj}}) \tag{9}$$

where $\lambda_{\mathrm{latent}}$ and $\lambda_{\mathrm{traj}}$ are hyperparameters controlling the regularization strength, and $s_t$ is an adaptive weighting term. For more details, refer to Sec. A.1.

Table 1: Quantitative extrapolation results on D-NeRF dataset. Metrics reported include PSNR, SSIM, and LPIPS-vgg. Best metric is highlighted in red, and second best in orange.

| Method | Lego | | | Mutant | | | Standup | | | Trex | | |
|---|---|---|---|---|---|---|---|---|---|---|---|---|
| | PSNR(↑) | SSIM(↑) | LPIPS(↓) | PSNR(↑) | SSIM(↑) | LPIPS(↓) | PSNR(↑) | SSIM(↑) | LPIPS(↓) | PSNR(↑) | SSIM(↑) | LPIPS(↓) |
| TiNeuVox-B | 23.34 | .9102 | .0942 | 24.40 | .9282 | .0700 | 21.77 | .9169 | .0927 | 20.72 | .9284 | .0751 |
| 4D-GS | 24.25 | .9150 | .0810 | 22.48 | .9300 | .0520 | 18.61 | .9180 | .0840 | 23.83 | .9460 | .0510 |
| Deformable-GS | 23.25 | .9349 | .0579 | 24.45 | .9310 | .0461 | 21.37 | .9124 | .0844 | 20.74 | .9421 | .0465 |
| GaussianPredict | 12.25 | .7594 | .2325 | 27.12 | .9514 | .0285 | 26.91 | .9456 | .0465 | 21.52 | .9443 | .0437 |
| 4D-Rotor-Gaussians | 22.32 | .9178 | .0705 | 30.62 | .9686 | .0252 | 25.79 | .9305 | .0632 | 20.21 | .9425 | .0692 |
| **Ours** | 25.74 | .9378 | .0547 | 34.53 | .9804 | .0126 | 28.91 | .9557 | .0360 | 22.04 | .9475 | .0485 |

| Method | Jumpingjacks | | | Bouncingballs | | | Hellwarrior | | | Hook | | |
|---|---|---|---|---|---|---|---|---|---|---|---|---|
| | PSNR(↑) | SSIM(↑) | LPIPS(↓) | PSNR(↑) | SSIM(↑) | LPIPS(↓) | PSNR(↑) | SSIM(↑) | LPIPS(↓) | PSNR(↑) | SSIM(↑) | LPIPS(↓) |
| TiNeuVox-B | 19.87 | .9115 | .0954 | 25.92 | .9677 | .0853 | 29.36 | .9097 | .1138 | 21.05 | .8817 | .1033 |
| 4D-GS | 19.95 | .9270 | .0770 | 29.55 | .9790 | .0340 | 16.84 | .8790 | .1250 | 22.03 | .9090 | .0670 |
| Deformable-GS | 20.32 | .9162 | .0790 | 29.49 | .9804 | .0237 | 30.15 | .9172 | .0799 | 21.60 | .8876 | .0820 |
| GaussianPredict | 20.12 | .9150 | .0811 | 28.09 | .9759 | .0322 | 30.75 | .9264 | .0767 | 23.75 | .9112 | .0553 |
| 4D-Rotor-Gaussians | 20.93 | .9063 | .1007 | 24.05 | .9401 | .0731 | 28.97 | .9006 | .1252 | 23.56 | .9156 | .0729 |
| **Ours** | 22.18 | .9243 | .0715 | 24.91 | .9660 | .0472 | 31.80 | .9365 | .0686 | 28.33 | .9493 | .0343 |

Table 2: Quantitative extrapolation results on NVFi dataset scenes. Metrics reported include PSNR, SSIM, and LPIPS. The best metric is highlighted in red, second-best is highlighted in orange.

| Method | fan | | | whale | | | shark | | | bat | | | telescope | | |
|---|---|---|---|---|---|---|---|---|---|---|---|---|---|---|---|
| | PSNR(↑) | SSIM(↑) | LPIPS(↓) | PSNR(↑) | SSIM(↑) | LPIPS(↓) | PSNR(↑) | SSIM(↑) | LPIPS(↓) | PSNR(↑) | SSIM(↑) | LPIPS(↓) | PSNR(↑) | SSIM(↑) | LPIPS(↓) |
| TiNeuVox | 26.91 | .9315 | .0643 | 27.20 | .9430 | .0579 | 30.95 | .9656 | .0367 | 28.65 | .9434 | .0663 | 27.04 | .9297 | .0507 |
| Deformable-GS | 23.75 | .9274 | .0519 | 26.58 | .9605 | .0386 | 29.11 | .9672 | .0273 | 27.07 | .9456 | .0482 | 22.92 | .9346 | .0459 |
| 4D-GS | 24.78 | .9565 | .0417 | 22.31 | .9638 | .0370 | 22.56 | .9648 | .0333 | 19.83 | .9565 | .0496 | 22.77 | .9414 | .0432 |
| GaussianPredict | 30.21 | .9682 | .0324 | 25.11 | .9610 | .0442 | 29.91 | .9695 | .0295 | 22.96 | .9587 | .0761 | 21.94 | .9381 | .0453 |
| NVFi | 27.17 | .9630 | .0370 | 26.03 | .9780 | .0290 | 28.87 | .9820 | .0210 | 25.02 | .9680 | .0420 | 27.10 | .9630 | .0460 |
| **Ours** | 33.49 | .9711 | .0303 | 33.86 | .9859 | .0135 | 38.73 | .9892 | .0082 | 36.68 | .9825 | .0176 | 36.57 | .9884 | .0057 |

| Method | fallingball | | | chessboard | | | darkroom | | | dining | | | factory | | |
|---|---|---|---|---|---|---|---|---|---|---|---|---|---|---|---|
| | PSNR(↑) | SSIM(↑) | LPIPS(↓) | PSNR(↑) | SSIM(↑) | LPIPS(↓) | PSNR(↑) | SSIM(↑) | LPIPS(↓) | PSNR(↑) | SSIM(↑) | LPIPS(↓) | PSNR(↑) | SSIM(↑) | LPIPS(↓) |
| TiNeuVox | 30.00 | .9500 | .0400 | 21.76 | .7567 | .2421 | 24.01 | .7400 | .1813 | 23.56 | .8443 | .1288 | 25.36 | .8222 | .1372 |
| Deformable-GS | 24.50 | .9200 | .0600 | 20.28 | .7866 | .2227 | 22.56 | .7423 | .2232 | 20.99 | .7922 | .2168 | 23.63 | .8107 | .1924 |
| 4D-GS | 22.00 | .9100 | .0700 | 20.71 | .8199 | .3444 | 21.99 | .7375 | .4036 | 22.08 | .8499 | .2800 | 23.42 | .8252 | .3356 |
| GaussianPredict | 21.50 | .9000 | .0800 | 20.12 | .7283 | .3168 | 20.01 | .6371 | .3840 | 18.01 | .6845 | .3811 | 21.11 | .8390 | .2708 |
| NVFi | 31.37 | .9780 | .0410 | 27.84 | .8720 | .2100 | 30.41 | .8260 | .2730 | 29.01 | .8980 | .1710 | 31.72 | .9080 | .1540 |
| **Ours** | 22.62 | .9244 | .0684 | 33.38 | .9266 | .0976 | 34.16 | .9019 | .1155 | 30.30 | .8829 | .1451 | 34.53 | .9183 | .1006 |

## 4 EXPERIMENTS

In this section, we evaluate ODE-GS for extrapolating dynamic 3D scenes. We present quantitative results in Sec. 4.1 to assess rendering quality on unseen future timestamps across D-NeRF (Pumarola et al., 2021), NVFi (Li et al., 2023), and HyperNeRF (Park et al., 2021b), and qualitative evaluations in Sec. 4.2 to demonstrate quality of the extrapolated dynamics through visual coherence.

### 4.1 QUANTITATIVE RESULTS

**D-NeRF.** On the D-NeRF dataset (Table 1), ODE-GS consistently surpasses both interpolation-based baselines (Deformable-GS, 4D-GS, 4D-Rotor-Gaussians) and extrapolation-oriented methods (GaussianPredict). Our model averages 27.30 Peak Signal-to-Noise Ratio (PSNR), 0.9497 SSIM, and 0.0467 LPIPS-vgg (Zhang et al., 2018), increasing metric performance against previous SOTA GaussianPrediction by 18.6%. We have especially large margins in Mutant (+10 dB) and Standup (+7 dB), where scene motion are very smooth and follow a straightforward trajectory. These gains directly validate our motivation: interpolation methods degrade when extrapolated beyond the training window due to their timestamp-conditioned design, while our Latent ODE approach enables extrapolation.

**NVFi.** The NVFi benchmark (Table 2) emphasizes robustness under both controlled single-object motion (fan, shark, telescope) and complex multi-object indoor dynamics (factory, darkroom, chessboard). ODE-GS achieves new state of the art across nearly all sequences, averaging 33.43 PSNR, 0.9471 SSIM, and 0.0603 LPIPS, improving previous SOTA method NVFi by 20%, and GaussianPrediction by 39.6% on averaging across scenes and metrics. Notably, in cluttered or occluded settings like factory and darkroom, our model reduces perceptual error (LPIPS) by more than 40% compared to baselines, demonstrating the benefit of decoupling reconstruction from forecasting.

**HyperNeRF.** On real-world HyperNeRF scenes (Table 3), ODE-GS establishes consistent improvements over baselines despite the higher noise and irregular dynamics of captured videos. Our method delivers lower LPIPS in split-cookie, slice-banana, and cut-lemon, and outperforms in both PSNR

Table 3: Quantitative extrapolation results on HyperNeRF. Metrics reported include PSNR, SSIM, and LPIPS. The best metric is highlighted in bold, second-best is underlined.

| Method | split-cookie | | | slice-banana | | | cut-lemon | | |
|---|---|---|---|---|---|---|---|---|---|
| | PSNR(↑) | SSIM(↑) | LPIPS(↓) | PSNR(↑) | SSIM(↑) | LPIPS(↓) | PSNR(↑) | SSIM(↑) | LPIPS(↓) |
| TiNeuVox | 16.67 | .6135 | .4778 | 18.44 | .6242 | .6119 | 18.84 | .6228 | .5743 |
| Deformable-GS | 17.84 | .5698 | .2945 | 21.73 | .6530 | .3241 | 21.36 | .6950 | .3207 |
| GaussianPredict | 16.93 | .5604 | .3336 | 21.97 | .6110 | .3749 | 20.91 | .6137 | .3220 |
| **Ours** | 20.72 | .6593 | .2406 | 21.29 | .6437 | .3230 | 21.69 | .6964 | .3098 |

| Method | keyboard | | | 3dprinter | | | chickchicken | | |
|---|---|---|---|---|---|---|---|---|---|
| | PSNR(↑) | SSIM(↑) | LPIPS(↓) | PSNR(↑) | SSIM(↑) | LPIPS(↓) | PSNR(↑) | SSIM(↑) | LPIPS(↓) |
| TiNeuVox | 19.03 | .6823 | .4665 | 18.16 | .5878 | .4949 | 15.23 | .6438 | .5888 |
| Deformable-GS | 19.98 | .6957 | .2497 | 19.89 | .6775 | .2378 | 19.03 | .6996 | .3167 |
| GaussianPredict | 20.13 | .6999 | .2511 | 19.96 | .6468 | .2209 | 21.94 | .6903 | .3284 |
| **Ours** | 21.06 | .7399 | .2327 | 20.22 | .6946 | .2242 | 20.29 | .7254 | .3023 |

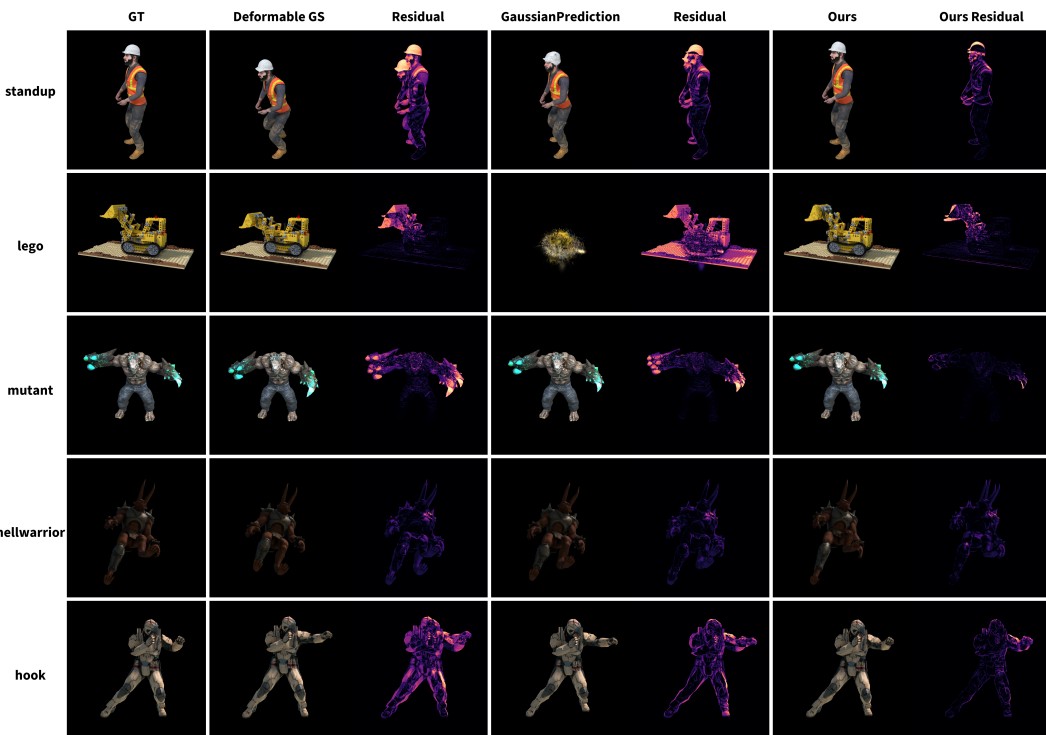

Figure 3: Qualitative visualization on 5 scenes from DNeRF dataset, from left to right are the ground truth image, rendered result from Deformable GS(Yang et al., 2024), residual of Deformable GS against GT, GaussianPrediction(Zhao et al., 2024), residual of GaussianPrediction against GT, and finally Our as well as Ours residual against GT.

and SSIM in chickchicken. These results show that ODE-GS avoids overfitting to noisy timestamp signals and remains stable under real-world non-idealities.

## 4.2 QUALITATIVE RESULTS

To provide a more intuitive understanding of our model's performance, we present a qualitative analysis of the extrapolated renderings, shown by Figure 3, which offers a compelling side-by-side comparison on five scenes from the D-NeRF dataset. While all models are tasked with predicting the scene at a future, unseen timestamp, the results vary dramatically. We show the difference in predicted motion via residual error maps, which visualize the pixel-wise error between the rendered images and the ground truth, where brighter regions indicate high error and dark regions indicate low error.

The error maps for both baselines show bright, structured residuals concentrated around the central object or character, revealing substantial inaccuracies in both shape and position. Conversely, the residual map for ODE-GS is significantly darker and less structured, providing clear visual evidence of a much lower prediction error. This demonstrates that by learning the underlying dynamics in a continuous latent space, ODE-GS not only preserves the high-frequency details but also forecasts motion more accurately for photorealistic novel-view synthesis in future frames. In particular, the scene Lego is recognized to have inaccurate poses by previous studies (Zhao et al., 2024; Yang et al., 2024), but our method can still extrapolate the scene that matches the ground truth image with low error, while methods like Gaussian Prediction (Zhao et al., 2024) can be less robust to such noises and fail completely at extrapolating the scene. For additional qualitative on NVFi see Figure 6.

## 4.3 ABLATION STUDY

We compare ODE-GS against four different abla­tion settings: Without ODE formulation (pure au­toregressive Transformer baseline trained under the same input–output supervision), without regulariza­tions, without adaptive regularization scaling, and without dynamic sampling. The results are summa­rized through average metrics in Table 4. In gen­eral, our full method outperforms all ablation settings. For our ablation setting without ODE, we use the Transformer encoder and decoder in a pure autore­gressive style. Specifically, this model variation has the same architecture only without the neural ODE. However, the autoregressive Transformer directly pre­dicts the next Gaussian parameters in discrete fixed steps, as each predicted output is then fed back as input for subsequent predictions. As shown in Ta­ble 4, the autoregressive baseline significantly under­performs, almost doubling the LPIPS metric. This highlights a limitation of discrete autoregressive mod­els also discussed in previous works (Chen et al., 2018; Rubanova et al., 2019): they lack the inherent smoothness prior naturally expressed by the ODE formulation. Therefore, abrupt jumps or oscillations in the predicted dynamics can occur. For the setting without regularization, we used only the extrapola­tion loss $L_{\mathbf{e}}$ for training objective. As shown on Table 4, the additional regularization narrows down solu­tion space further, which results in higher average metrics. In Figure 4, we observe these additional constraints helps the model improve on scenes with more complex and diverse motion such as the dining and hell-warrior scene. For per-scene results, refer to Table 8 in the Appendix.

Table 4: Ablation study average results over the NVFi dataset.

| Method | PSNR($\uparrow$) | SSIM($\uparrow$) | LPIPS($\downarrow$) |
|---|---|---|---|
| w/o ODE | 23.71 | .879 | .113 |
| w/o Regularization | 32.90 | .943 | .066 |
| w/o Adaptive reg. | 32.19 | .938 | .068 |
| w/o Dynamic sampling | 31.35 | .935 | .069 |
| **Ours** | **33.43** | **.947** | **.060** |

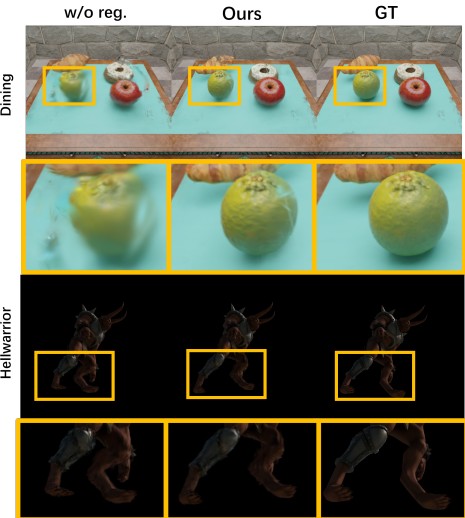

Figure 4: Qualitative comparison on ODE-GS trained using latent and trajectory regulariza­tion vs. using only extrapolation loss on two selected scenes. We highlight the areas within each scene with highest visual disparity.

## 4.4 ADDITIONAL EXPERIMENTS

**Reprojection loss joint training** We have experimented with adding the image reprojection loss to the training of the extrapolation model, shown in Table 5. Specifically, for every iteration, we sample a random camera with ground truth image, and project the gaussians onto the image via the rendering pipeline (Sec. 3). We follow the original 3DGS (Kerbl et al., 2023) and apply L1+SSIM loss. As shown in Table 5, the projection loss does not provide any significant metric improvement over our base model. This is likely due to the interpolation model being trained using the projection loss and having already accurately represented the training data. Thus, trajectory loss itself is already sufficient as supervision.

Table 5: The effect of additional reprojection loss during training on D-NeRF extrapolation. Metrics reported include PSNR, SSIM, and LPIPS-vgg.

| Method | Lego | | | Mutant | | | Standup | | | Trex | | |
|---|---|---|---|---|---|---|---|---|---|---|---|---|
| | PSNR(↑) | SSIM(↑) | LPIPS(↓) | PSNR(↑) | SSIM(↑) | LPIPS(↓) | PSNR(↑) | SSIM(↑) | LPIPS(↓) | PSNR(↑) | SSIM(↑) | LPIPS(↓) |
| **Ours** | 25.74 | .9378 | .0547 | 34.53 | .9804 | .0126 | 28.91 | .9557 | .0360 | 22.04 | .9475 | .0485 |
| **Ours with reprojection** | 25.63 | .9368 | .0549 | 29.72 | .9671 | .0219 | 29.27 | .9546 | .0380 | 22.02 | .9469 | .0496 |
| Method | Jumpingjacks | | | Bouncingballs | | | Hellwarrior | | | Hook | | |
| | PSNR(↑) | SSIM(↑) | LPIPS(↓) | PSNR(↑) | SSIM(↑) | LPIPS(↓) | PSNR(↑) | SSIM(↑) | LPIPS(↓) | PSNR(↑) | SSIM(↑) | LPIPS(↓) |
| **Ours** | 22.18 | .9243 | .0715 | 24.91 | .9660 | .0472 | 31.80 | .9365 | .0686 | 28.33 | .9493 | .0343 |
| **Ours with reprojection** | 21.32 | .9205 | .0758 | 25.96 | .9684 | .0436 | 31.48 | .9331 | .0704 | 27.80 | .9454 | .0368 |

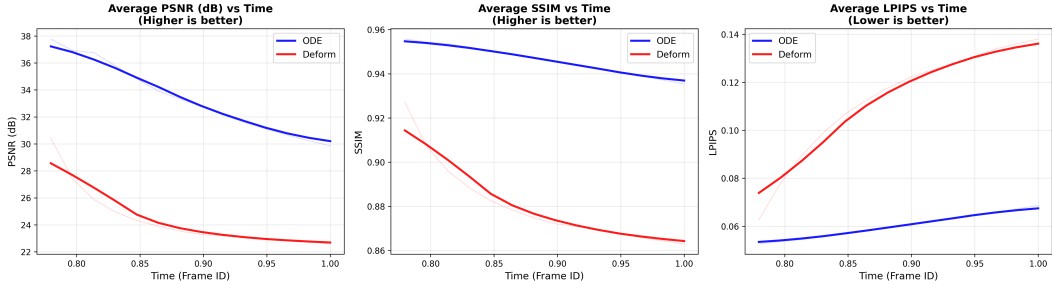

Figure 5: Visualization of the degrade in metrics through time for extrapolation. Y axis is the corresponding metric and X axis is the time or frame where the metric was evaluated. This graph is the average over all scenes in NVFi dataset.

**Performance degrade analysis** Shown in Figure 5 as well as per-scene figure 8,7,11,12,14,10,15,13, we analyzed the performance degrade of the rendered results during extrapolation period. On the NVFi dataset, our model not only shows an overall high metric performance, but also a slower degrade rate in time. Especially the SSIM and LPIPS measures degrade much slower than the baseline Yang et al. (2024), showing our method's capability of maintaining plausible motion and object appearance as extrapolation length becomes longer.

**Effect of different input time spans** On certain scenes, the model can benefit from a shorter input context time span $T_c$. Specifically, we reduced time span from $0.6$ to $0.1$ on various train/test time splits over the bouncingballs scene, ranging from using 60 percent of temporal duration for training to 95 percent, shown in Table 11 in the appendix. Compared to default setting, our method achieves a $21.70\%$ average improvement across the three metrics over previous best method in Table 1.

## 5 LIMITATIONS AND CONCLUSION

Our model inherits the quality of the underlying interpolation model used to generate Gaussian trajectories. If the interpolation stage fails to accurately reconstruct the scene within the observed window (for instance, in cases of fast-moving small objects like the *fallingball* scene) then the subsequent extrapolation will propagate these errors forward. In scenarios like *bouncingballs* and *trex* where the evolution of the scene undergoes abrupt changes, discontinuities, or fundamentally novel behaviors not foreshadowed by the past, the model's predictions degrade. One possible way to address this limitation is to utilize data-driven priors for extrapolation that can generalize across scenes instead of overfitting to the observed dynamics of specific scenes. We have conducted a preliminary experiment on our model's generalization capability across scenes B.2.

In summary, we introduce ODE-GS, which integrates 3D Gaussian Splatting with Transformer-based latent neural ODEs to achieve dynamic scene extrapolation. By decoupling scene reconstruction from temporal forecasting, enforcing smoothness through continuous-time latent dynamics, and incorporating adaptive regularization and dynamic sampling, ODE-GS achieves state-of-the-art on synthetic (D-NeRF, NVFi) and real-world (HyperNeRF) benchmarks. We show that modeling scene evolution as latent flows mitigates out-of-distribution failures common in timestamp-conditioned methods and enables accurate extrapolation of scene dynamics beyond the observed time window.

ACKNOWLEDGMENTS

This work was supported by NSF 2112562 Athena AI Institute, NSF Graduate Research Fellowships Program (GRFP), and the Yale AI Seed Grant 2025

ETHICS STATEMENT

This work advances methods for forecasting dynamics in 3D scenes using Gaussian splatting and latent ODEs. Given the scope of the work, we do not identify immediate ethical risks associated with the approach itself. However, as with any machine learning system, outcomes depend on the quality and diversity of the training data. If deployed in downstream applications such as robotics or autonomous navigation, inaccurate forecasts of scene dynamics may pose safety risks, and care should be taken to validate performance in those contexts.

REPRODUCIBILITY STATEMENT

We provide detailed descriptions of our approach in Sec. 3, including architecture design, sampling strategies, and training objectives. Experimental settings, dataset usage, and evaluation metrics are reported in Sec. B.1, with additional implementation details included in Appendix B. To ensure reproducibility, we have released the implementation of this paper publicly online.

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

## A APPENDIX

### A.1 DETAILS ON ADAPTIVE WEIGHTING FOR REGULARIZATION

This mechanism operates at each training iteration by using the Exponential Moving Average (EMA) of the trajectory prediction loss as a proxy for the model's convergence state. The regularization weight increased as the extrapolation loss decreases, allowing the regularizers to more strongly guide the final, fine-grained trajectory predictions.

The scaling factor $s_t$ at iteration $t$ is computed as:

$$s_t = \exp\left(-\frac{1}{\tau} \cdot \text{clip}\left(\frac{\mathcal{L}_{\text{EMA}}(t) - \mathcal{L}_{\text{end}}}{\mathcal{L}_{\text{init}} - \mathcal{L}_{\text{end}}}, 0, 1\right)\right) \tag{10}$$

where:

- $\mathcal{L}_{\text{EMA}}(t)$ is the EMA of the trajectory prediction loss at iteration $t$.
- $\mathcal{L}_{\text{init}}$ and $\mathcal{L}_{\text{end}}$ are hyperparameters representing the expected initial and final extrapolation loss values.
- $\tau$ is a temperature hyperparameter that controls the decay rate of the scaling. A lower $\tau$ leads to a faster decay and a more aggressive increase in regularization.
- The 'clip' function normalizes the loss into the range $[0, 1]$, ensuring the scaling factor remains between $(0, 1]$.

The EMA of the trajectory prediction loss, $\mathcal{L}_{\text{EMA}}(t)$, is calculated as:

$$\mathcal{L}_{\text{EMA}}(t) = \alpha \cdot \mathcal{L}_{\text{EMA}}(t - 1) + (1 - \alpha) \cdot \mathcal{L}_{\text{e}}(t) \tag{11}$$

where $\mathcal{L}_{\text{e}}(t)$ is the extrapolation loss at the current iteration and $\alpha$ is the EMA decay rate. The final regularization weights for the latent and trajectory regularizers are then scaled by $s_t$ at each iteration before being added to the total loss.

### A.2 PROBABILISTIC FORECASTING WITH A VARIATIONAL LATENT ODE

Many past Latent ODE works that focus on extrapolation and forecasting has been using a variational formulation for the model (Rubanova et al., 2019). Specifically, they follow the Variational Autoencoder (Kingma & Welling, 2013) approach. We provide details on our model variant which uses this approach instead of deterministic modeling, as well as quantitative results compare the two methods on the NVFi dataset, as shown in table 6

**Variational Variant for the Transformer ODE architecture.** Instead of mapping the observed history to a single initial state for the ODE, prior works often formulate trajectory forecasting as a probabilistic problem (Chen et al., 2018). Given a historical trajectory $\gamma_k = \{G_k(t_j)\}_{j=1}^{N_c}$ for a Gaussian $k$, the Transformer encoder produces a latent vector $h_k$. A projection head then parameterizes a Gaussian posterior distribution over the initial latent state,

$$q_\phi\big(z_k(t_0) \mid \gamma_k\big) = \mathcal{N}\big(\mu_{z_k}, \text{diag}(\sigma_{z_k}^2)\big), \tag{12}$$

where $z_k(t_0)$ is the latent state at the start of extrapolation. During training, we sample from this distribution via the reparameterization trick:

$$z_k(t_0) = \mu_{z_k} + \sigma_{z_k} \odot \epsilon, \qquad \epsilon \sim \mathcal{N}(0, I). \tag{13}$$

The sampled latent state $z_k(t_0)$ is then evolved forward in time by numerically solving the latent ODE,

$$z_k(t) = \texttt{ODESolve}\big(f_\theta, z_k(t_0), t\big), \tag{14}$$

and subsequently decoded to predicted Gaussian parameters

$$\hat{G}_k(t) = \delta_\psi(z_k(t)). \tag{15}$$

Table 6: Comparison between our deterministic and variational formulations on NVFi dataset scenes. Metrics reported include PSNR, SSIM, and LPIPS.

| Scene | Ours (Deterministic) | | | Ours (Variational) | | |
|---|---|---|---|---|---|---|
| | PSNR($\uparrow$) | SSIM($\uparrow$) | LPIPS($\downarrow$) | PSNR($\uparrow$) | SSIM($\uparrow$) | LPIPS($\downarrow$) |
| factory | 34.53 | .9183 | .1006 | 22.57 | .7851 | .218 |
| dining | 30.30 | .8829 | .1451 | 16.19 | .3981 | .595 |
| darkroom | 34.16 | .9019 | .1155 | 22.83 | .7098 | .233 |
| whale | 33.86 | .9859 | .0135 | 26.49 | .9601 | .038 |
| shark | 38.73 | .9892 | .0082 | 29.43 | .9652 | .030 |
| chessboard | 33.38 | .9266 | .0976 | 19.69 | .7667 | .244 |
| bat | 36.68 | .9825 | .0176 | 28.01 | .9554 | .043 |
| fan | 33.49 | .9711 | .0303 | 21.67 | .8589 | .114 |
| telescope | 36.57 | .9884 | .0057 | 22.66 | .9283 | .053 |
| fallingball | 22.62 | .9244 | .0684 | 9.45 | .7559 | .206 |

**Objective.** The variational model is trained by maximizing the Evidence Lower Bound (ELBO), which corresponds to minimizing

$$\mathcal{L}_{\text{var-e}} = \sum_{t \in \mathcal{T}_e} \Big[ \underbrace{- \log p_\sigma\big(G_k(t) \mid \hat{G}_k(t)\big)}_{\text{prediction NLL}} + \text{KL}\Big[ q_\phi\big(z_k(t_0) \mid \gamma_k\big) \,\|\, p\big(z_k(t_0)\big) \Big] \Big] \quad (16)$$

where $\mathcal{T}_e$ denotes the extrapolation timestamps. The first term encourages predicted Gaussian trajectories $\hat{G}_k(t)$ to align with ground truth $G_k(t)$, the second regularizes the latent space toward a unit Gaussian prior $p(z_k(t_0))$. The final objective is then to compose this loss with regularization as discussed in 3.4.

While variational formulations provide a principled way to capture uncertainty in trajectory forecasting, our results 6 show that the deterministic version of ODE-GS significantly outperforms its variational counterpart across all NVFi scenes. This gap arises because variational training introduces additional complexity through posterior sampling and KL regularization, which can destabilize optimization when data is limited and the ground-truth dynamics are relatively deterministic. In practice, the model tends to underfit sharp motion patterns, producing blurred or averaged predictions that reduce both PSNR and SSIM while inflating perceptual error (LPIPS). By contrast, the deterministic formulation directly learns smooth latent flows aligned with observed trajectories, avoiding posterior collapse and better exploiting the strong spatio-temporal smoothness priors inherent in dynamic 3D scenes.

## B  DETAILS ON DYNAMIC TRAJECTORY SAMPLING

We provide additional information on our sampling strategy introduced in Section 3.3. Training data for the extrapolation module are derived from the pre-trained interpolation model, which can generate Gaussian trajectories at arbitrary real-valued timestamps through the deformation function $\mathcal{D}_\omega$.

Each training sample is indexed by a Gaussian $k$ and a starting time $t_0$. We fix the number of context steps $N_c$, target steps $N_e$, and the context time span $T_c$.

The context sequence is uniformly sampled as

$$\Delta_c = \frac{T_c}{N_c - 1}, \qquad \gamma_c^{(i)} = \big\{ G_k(t_0 + i\Delta_c) \big\}_{i=1}^{N_c}. \quad (17)$$

The target sequence begins immediately after the context window at $t_c^{\text{end}} = t_0 + T_c$ and spans the remaining horizon:

$$T_e = t_{\max} - t_c^{\text{end}}, \qquad \Delta_e = \frac{T_e}{N_e}, \qquad \gamma_e^{(i)} = \big\{ G_k(t_c^{\text{end}} + i\Delta_e) \big\}_{i=1}^{N_e}. \quad (18)$$

This ensures $N_c$ and $N_e$ remain fixed across all samples, while $T_e$ varies with $t_0$. The variability in target length naturally generates prediction tasks of differing difficulty, ranging from short- to long-term forecasts.

Table 7: Per-scene ablation over NVFi of our full method against removal of ODE and removal of dynamic trajectory sampling. Metrics reported include PSNR, SSIM, and LPIPS.

| Scene | w/o ODE | | | w/o dynamic trajectory sampling | | | Ours | | |
|---|---|---|---|---|---|---|---|---|---|
| | PSNR($\uparrow$) | SSIM($\uparrow$) | LPIPS($\downarrow$) | PSNR($\uparrow$) | SSIM($\uparrow$) | LPIPS($\downarrow$) | PSNR($\uparrow$) | SSIM($\uparrow$) | LPIPS($\downarrow$) |
| factory | 24.06 | .814 | .179 | 33.84 | .915 | .102 | 34.53 | .918 | .101 |
| dining | 20.98 | .784 | .217 | 25.47 | .810 | .204 | 30.30 | .883 | .145 |
| darkroom | 22.08 | .731 | .227 | 33.89 | .900 | .117 | 34.16 | .902 | .116 |
| chessboard | 19.84 | .784 | .225 | 32.27 | .920 | .100 | 33.38 | .927 | .098 |
| bat | 28.42 | .960 | .039 | 35.37 | .979 | .023 | 36.68 | .983 | .018 |
| telescope | 23.20 | .937 | .040 | 33.09 | .983 | .010 | 36.57 | .988 | .006 |
| fallingball | 17.14 | .904 | .094 | 26.22 | .936 | .064 | 22.62 | .924 | .068 |
| fan | 23.35 | .931 | .052 | 26.97 | .948 | .043 | 33.49 | .971 | .030 |
| shark | 29.47 | .967 | .026 | 33.17 | .980 | .015 | 38.73 | .989 | .008 |
| whale | 28.51 | .973 | .028 | 33.21 | .984 | .015 | 33.86 | .986 | .014 |
| **Average** | **23.71** | **.879** | **.113** | **31.35** | **.935** | **.069** | **33.43** | **.947** | **.060** |

The complete dataset is defined as the union over all Gaussian indices and valid starting times:

$$\mathcal{D} = \bigcup_{k=1}^{K} \bigcup_{t_0 \in \mathcal{T}_k} \left\{ (\gamma_c^{(i)}, \gamma_e^{(i)}) \right\}. \tag{19}$$

During training, the context $\gamma_c^{(i)}$ is processed by the Transformer encoder–ODE module to predict $\hat{\gamma}_e^{(i)} = \{\hat{G}_k(t)\}_{t \in T_e}$. The prediction is supervised with an $L_1$ extrapolation loss:

$$\mathcal{L}_e = \frac{1}{N_e} \sum_{j=1}^{N_e} \left\| \hat{G}_k(t_j) - G_k(t_j) \right\|_1. \tag{20}$$

At inference, we simply take the final context segment of length $N_c$ from the observed window and apply the same extrapolation procedure.

### B.1 IMPLEMENTATION DETAILS

**Model architecture.** The Transformer encoder has $d_{\text{model}} = 128$ latent dimensions with $n_{\text{head}} = 8$ attention heads and $L_{\text{enc}} = 5$ encoder layers. For the latent ODE component, we set the latent dimension to $d_{\text{latent}} = 64$ and use an MLP with $L_{\text{ode}} = 4$ layers and $d_{\text{hidden}} = 64$ hidden units per layer. The decoder network mirrors this structure with $L_{\text{dec}} = 5$ layers and $d_{\text{hidden}} = 128$ hidden units. For the ODE function, we use Tanh activations for smooth and bounded outputs. For the interpolation model, we follow Deformable GS (Yang et al., 2024) implementation.

**Hyperparameters.** Our two-stage training pipeline is implemented in PyTorch with the original 3D Gaussian Splatting renderer (Kerbl et al., 2023). For the interpolation stage, we set the learning rate to $8 \times 10^{-4}$ with a cosine annealing scheduler that decays to a minimum of $1.6 \times 10^{-6}$, training for 40k iterations on images with timestamps prior to our dataset-specific train/test split (Sec. B.2.1).

For the extrapolation stage, we solve latent ODEs using the `torchode` package (Kidger et al., 2021) with the adaptive DOPRI5 solver configured with tolerances $rtol = 10^{-3}$ and $atol = 10^{-4}$. We train on NVIDIA GPUs (RTX 3090 or A6000) with a batch size of 512 for 40 epochs, using an initial learning rate of $1 \times 10^{-3}$ and cosine annealing down to $1 \times 10^{-6}$. The input context sequence length is set to $N_c = 30$ and the extrapolation length to $N_e = 10$ during optimization. For adaptive trajectory sampling, the temperature parameter is set to $0.05$ on D-NeRF and $0.15$ on HyperNeRF and NVFi. For adaptive regularization, we use $\mathcal{L}_{\text{init}} = 0.02$ for D-NeRF and NVFi and $0.05$ for HyperNeRF, we set $\mathcal{L}_{\text{end}} = 0.0$ over training on all datasets. Our EMA decay rate is set at $0.9$ We use $\lambda_{\text{traj}} = 10^{-1}$ for the trajectory regularizer and $\lambda_{\text{latent}} = 10^{-5}$ for the latent regularizer.

### B.2 MULTI-SCENE GENERALIZATION EXPERIMENT

We evaluate our extrapolation model's capability to generalize across scenes in Table 9. Specifically, we set up a mult-scene training experiment where we collect all generated trajectories from the

Table 8: Per-scene ablation over NVFi of our full method against removal of regularizers and removal of adaptive regularizer scaling. Metrics reported include PSNR, SSIM, and LPIPS.

| Scene | w/o regularizers | | | w/o adaptive regularizer scaling | | | Ours | | |
|---|---|---|---|---|---|---|---|---|---|
| | PSNR(↑) | SSIM(↑) | LPIPS(↓) | PSNR(↑) | SSIM(↑) | LPIPS(↓) | PSNR(↑) | SSIM(↑) | LPIPS(↓) |
| factory | 34.68 | .919 | .102 | 33.53 | .912 | .104 | 34.53 | .918 | .101 |
| dining | 27.26 | .841 | .180 | 25.99 | .828 | .187 | 30.30 | .883 | .145 |
| darkroom | 33.81 | .896 | .123 | 32.32 | .880 | .127 | 34.16 | .902 | .116 |
| chessboard | 33.16 | .929 | .096 | 32.16 | .921 | .099 | 33.38 | .927 | .098 |
| bat | 37.43 | .985 | .017 | 36.34 | .983 | .018 | 36.68 | .983 | .018 |
| telescope | 36.23 | .988 | .006 | 35.68 | .988 | .006 | 36.57 | .988 | .006 |
| fallingball | 19.35 | .918 | .086 | 19.53 | .918 | .085 | 22.62 | .924 | .068 |
| fan | 34.96 | .976 | .027 | 35.23 | .974 | .028 | 33.49 | .971 | .030 |
| shark | 38.26 | .989 | .009 | 36.63 | .986 | .010 | 38.73 | .989 | .008 |
| whale | 33.88 | .987 | .013 | 34.52 | .987 | .013 | 33.86 | .986 | .014 |
| **Average** | **32.90** | **.943** | **.066** | **32.19** | **.938** | **.068** | **33.43** | **.947** | **.060** |

respective interpolation models from each scene in the NVFi dataset, union them into the same dataset, and train one extrapolation model (The Transformer Latent ODE) on all trajectories simultaneously. This is equivalent to having the scenes share the same set of extrapolation model weights during training. To assess the generalization of the model, we intentionally hold out the *whale* scene's observed trajectories to be excluded from this dataset union. The last rows (Ours Multi) of Table 9 shows our quantitative results of extrapolation for each scene using this shared-weight model for extrapolation, conditioned on the last segment of Gaussian trajectories for each scene's unique interpolation model. Our quantitative numbers show that our extrapolation model can fit to multiple scene trajectories at the same time, while being capable of generalizing to unseen scene dynamics with simple motion patterns like *whale*. Although simultaneously training on multiple scene trajectories does not make the extrapolation model exceed metric performs compared to Our default per-scene training, it shows the potential of such and approach for future works to explore.

### B.2.1 DATASETS DETAILS

We evaluate ODE-GS on three datasets: the D-NeRF dataset (Pumarola et al., 2021), the NVFi dataset (Li et al., 2023), and the HyperNeRF dataset (Park et al., 2021b). The D-NeRF dataset comprises eight synthetic dynamic scenes (Lego, Mutant, Standup, Trex, Jumpingjacks, Bouncingballs, Hellwarrior, Hook), each containing 100-200 training images and 20 test images with timestamps normalized from 0 to 1, rendered at 800×800 resolution with black backgrounds. The NVFi dataset is also a synthetic dataset that provides two subcategories: the simpler Dynamic Object Dataset featuring rotating objects (fan, whale, shark, bat, telescope) and the more challenging Dynamic Indoor Scene Dataset (chessboard, darkroom, dining, factory) containing multi-object scenes with occlusions and realistic lighting variations. The HyperNeRF dataset is a series of monocular videos on day-to-day scenes with varying motion complexity. We use the split-cookie, slice-banan, cut-lemon, keyboard, 3dprinter, and chickchicken scenes, picked at random. For the D-Nerf dataset, we use 80% of the temporal sequence for training and reserve the final 20% as ground truth for extrapolation evaluation. For the HyperNeRF dataset, we use the first 90% of the temporal sequence. For the NVFi dataset, we follow the original train-test split where 75% of the temporal sequence is used for training and the other 25% is reserved for testing. Our model's context window time span is set differently for each dataset, with $T_c = 0.6$ for D-NeRF, $T_c = 0.5$ for NVFi and HyperNeRF.

### B.2.2 JOINT TRAINING EXPERIMENT

In order to justify our design of training the extrapolation model while freezing the interpolation model, we've tested our method with end-to-end training on both models simultaneously. Specifically, we keep the parameters of the interpolation model to be trainable while we train the extrapolation model, supervised by the same loss. As shown by Table 10, the overall performance decreases on all three metrics.

Table 9: Quantitative extrapolation results on NVFi dataset scenes for training the extrapolation model on multi-scene setting. Scene *whale* is not included in the training data. Metrics reported include PSNR, SSIM, and LPIPS. The best metric is highlighted in red, second-best is highlighted in orange.

| Method | fan | | | dining | | | factory | | |
|---|---|---|---|---|---|---|---|---|---|
| | PSNR(↑) | SSIM(↑) | LPIPS(↓) | PSNR(↑) | SSIM(↑) | LPIPS(↓) | PSNR(↑) | SSIM(↑) | LPIPS(↓) |
| TiNeuVox | 26.91 | .9315 | .0643 | 23.56 | .8443 | .1288 | 25.36 | .8222 | .1372 |
| Deformable-GS | 23.75 | .9274 | .0519 | 20.99 | .7922 | .2168 | 23.63 | .8107 | .1924 |
| 4D-GS | 24.78 | .9565 | .0417 | 22.08 | .8499 | .2800 | 23.42 | .8252 | .3356 |
| GaussianPredict | 30.21 | .9682 | .0324 | 18.01 | .6845 | .3811 | 21.11 | .8390 | .2708 |
| NVFi | 27.17 | .9630 | .0370 | 29.01 | .8980 | .1710 | 31.72 | .9080 | .1540 |
| Ours Det. | 33.49 | .9711 | .0303 | 30.30 | .8829 | .1451 | 34.53 | .9183 | .1006 |
| Ours Multi. | 27.30 | .9494 | .0422 | 27.61 | .8530 | .1591 | 31.84 | .8956 | .1110 |

| Method | shark | | | bat | | | telescope | | |
|---|---|---|---|---|---|---|---|---|---|
| | PSNR(↑) | SSIM(↑) | LPIPS(↓) | PSNR(↑) | SSIM(↑) | LPIPS(↓) | PSNR(↑) | SSIM(↑) | LPIPS(↓) |
| TiNeuVox | 30.95 | .9656 | .0367 | 28.65 | .9434 | .0663 | 27.04 | .9297 | .0507 |
| Deformable-GS | 29.11 | .9672 | .0273 | 27.07 | .9456 | .0482 | 22.92 | .9346 | .0459 |
| 4D-GS | 22.56 | .9648 | .0333 | 19.83 | .9565 | .0496 | 22.77 | .9414 | .0432 |
| GaussianPredict | 29.91 | .9695 | .0295 | 22.96 | .9587 | .0761 | 21.94 | .9381 | .0453 |
| NVFi | 28.87 | .9820 | .0210 | 25.02 | .9680 | .0420 | 27.10 | .9630 | .0460 |
| Ours | 38.73 | .9892 | .0082 | 36.68 | .9825 | .0176 | 36.57 | .9884 | .0057 |
| Ours Multi. | 37.31 | .9869 | .0094 | 36.72 | .9818 | .0188 | 34.19 | .9838 | .0085 |

| Method | fallingball | | | chessboard | | | darkroom | | |
|---|---|---|---|---|---|---|---|---|---|
| | PSNR(↑) | SSIM(↑) | LPIPS(↓) | PSNR(↑) | SSIM(↑) | LPIPS(↓) | PSNR(↑) | SSIM(↑) | LPIPS(↓) |
| TiNeuVox | 30.00 | .9500 | .0400 | 21.76 | .7567 | .2421 | 24.01 | .7400 | .1813 |
| Deformable-GS | 24.50 | .9200 | .0600 | 20.28 | .7866 | .2227 | 22.56 | .7423 | .2232 |
| 4D-GS | 22.00 | .9100 | .0700 | 20.71 | .8199 | .3444 | 21.99 | .7375 | .4036 |
| GaussianPredict | 21.50 | .9000 | .0800 | 20.12 | .7283 | .3168 | 20.01 | .6371 | .3840 |
| NVFi | 31.37 | .9780 | .0410 | 27.84 | .8720 | .2100 | 30.41 | .8260 | .2730 |
| Ours | 22.62 | .9244 | .0684 | 33.38 | .9266 | .0976 | 34.16 | .9019 | .1155 |
| Ours Multi | 23.54 | .9219 | .0779 | 29.51 | .8949 | .1116 | 29.92 | .8489 | .1364 |

| Method | whale | | | average | | | | | |
|---|---|---|---|---|---|---|---|---|---|
| | PSNR(↑) | SSIM(↑) | LPIPS(↓) | PSNR(↑) | SSIM(↑) | LPIPS(↓) | | | |
| TiNeuVox | 27.20 | .9430 | .0579 | 26.54 | .8826 | .1005 | | | |
| Deformable-GS | 26.58 | .9605 | .0386 | 24.14 | .8787 | .1127 | | | |
| 4D-GS | 22.31 | .9638 | .0370 | 22.25 | .8926 | .1638 | | | |
| GaussianPredict | 25.11 | .9610 | .0442 | 23.09 | .8584 | .1660 | | | |
| NVFi | 26.03 | .9780 | .0290 | 28.45 | .9336 | .1024 | | | |
| Ours | 33.86 | .9859 | .0135 | 33.43 | .9471 | .0603 | | | |
| Ours Multi. | 29.22 | .9693 | .0260 | 30.72 | .9285 | .0701 | | | |

Table 10: Comparison between per-scene (Ours) and joint training (Ours joint) on the D-NeRF dataset. Metrics reported include PSNR, SSIM, and LPIPS-vgg.

| Method | Metric | Lego | Mutant | Standup | Trex | Jumpingjacks | Bouncingballs | Hellwarrior | Hook | Avg |
|---|---|---|---|---|---|---|---|---|---|---|
| Ours | PSNR(↑) | 25.74 | 34.53 | 28.91 | 22.04 | 22.18 | 24.91 | 31.80 | 28.33 | 27.31 |
| Ours joint | PSNR(↑) | 24.93 | 21.10 | 20.82 | 21.19 | 19.62 | 19.60 | 27.75 | 19.28 | 21.79 |
| Ours | SSIM(↑) | .9378 | .9804 | .9557 | .9475 | .9243 | .9660 | .9365 | .9493 | .9497 |
| Ours joint | SSIM(↑) | .9363 | .8979 | .9097 | .9371 | .9121 | .9467 | .8948 | .8632 | .9122 |
| Ours | LPIPS(↓) | .0547 | .0126 | .0360 | .0485 | .0715 | .0472 | .0686 | .0343 | .0467 |
| Ours_joint | LPIPS(↓) | .0565 | .1144 | .1087 | .0663 | .1091 | .0894 | .1087 | .1339 | .0984 |

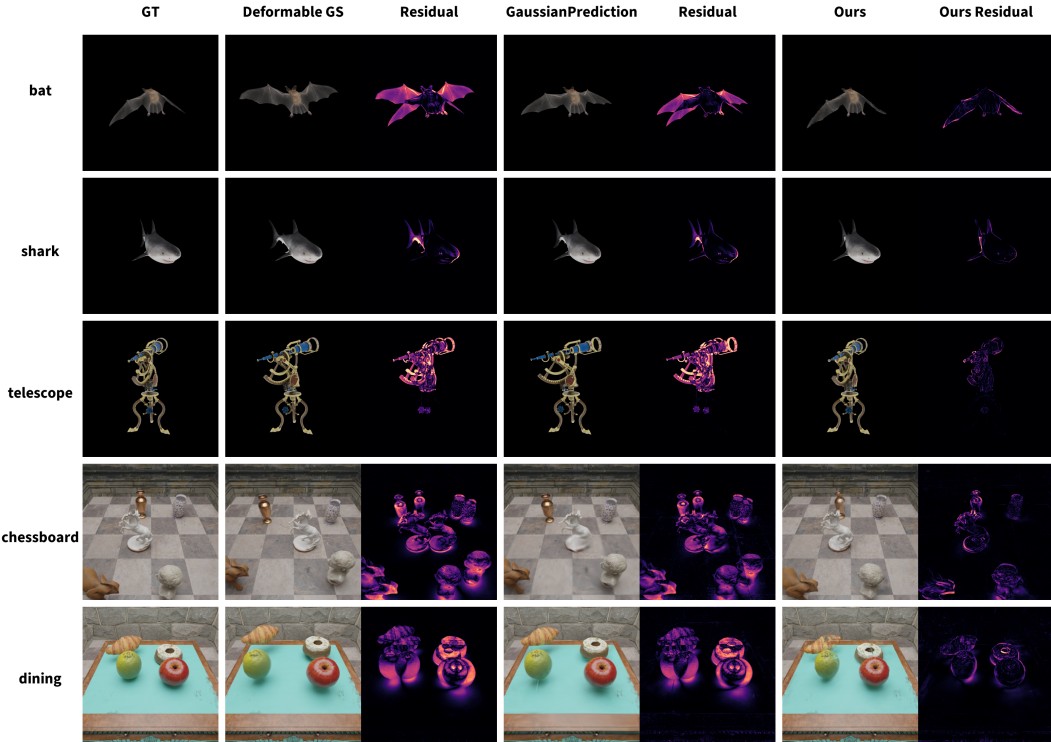

Figure 6: Qualitative results on 5 scenes from the NVFI (Li et al., 2023) dataset, from left to right are the ground truth image, rendered result from Deformable GS(Yang et al., 2024), residual of Deformable GS against GT, GaussianPrediction(Zhao et al., 2024), residual of GaussianPrediction against GT, and finally Our as well as Ours residual against GT.

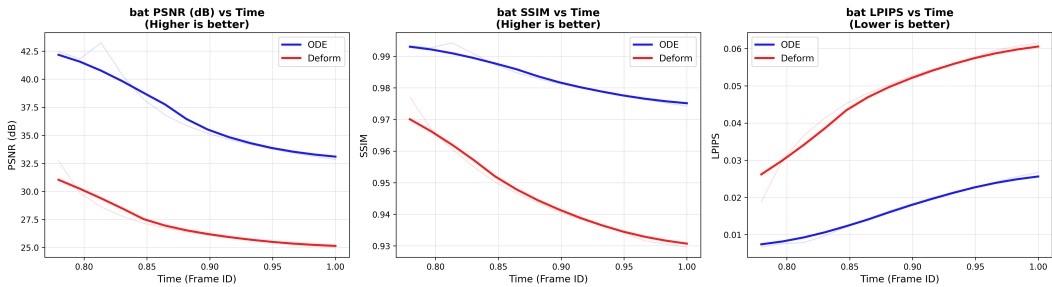

Figure 7: Metric performance on extrapolation over time compared with Deform-GS on scene bat from NVFi

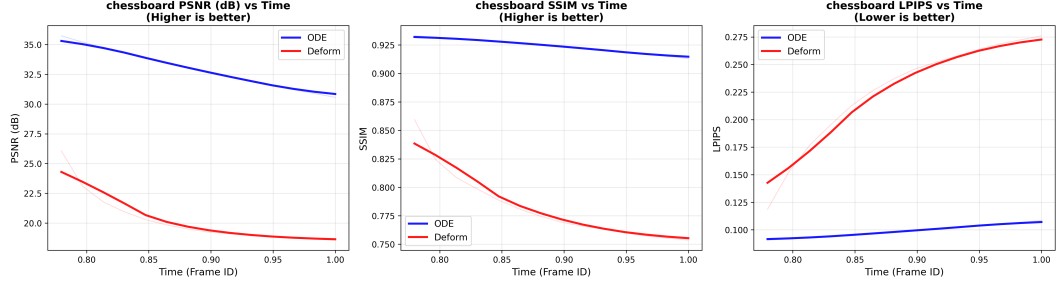

Figure 8: Metric performance on extrapolation over time compared with Deform-GS on scene chessboard from NVFi

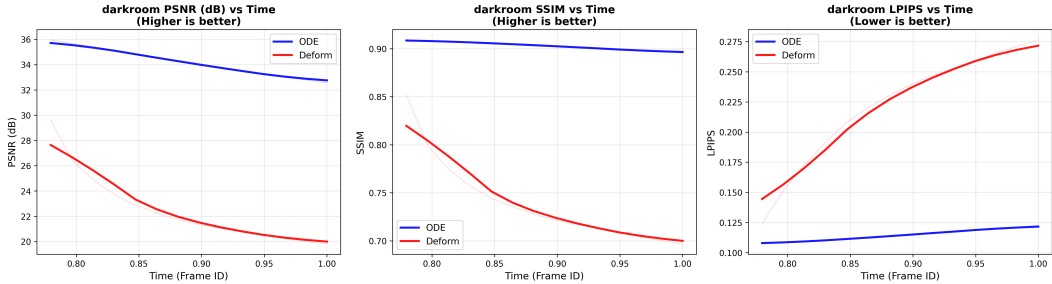

Figure 9: Metric performance on extrapolation over time compared with Deform-GS on scene bat from darkroom

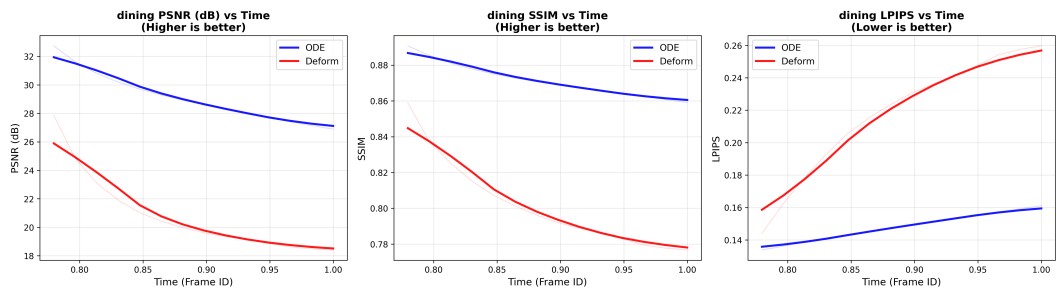

Figure 10: Metric performance on extrapolation over time compared with Deform-GS on scene dining from NVFi

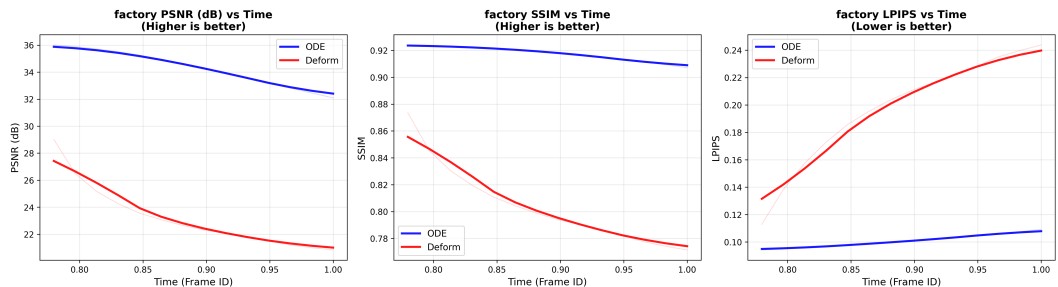

Figure 11: Metric performance on extrapolation over time compared with Deform-GS on scene factory from NVFi

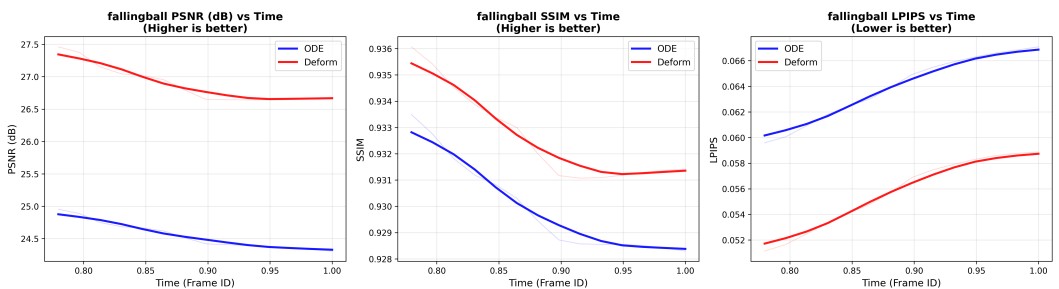

Figure 12: Metric performance on extrapolation over time compared with Deform-GS on scene falling ball from NVFi

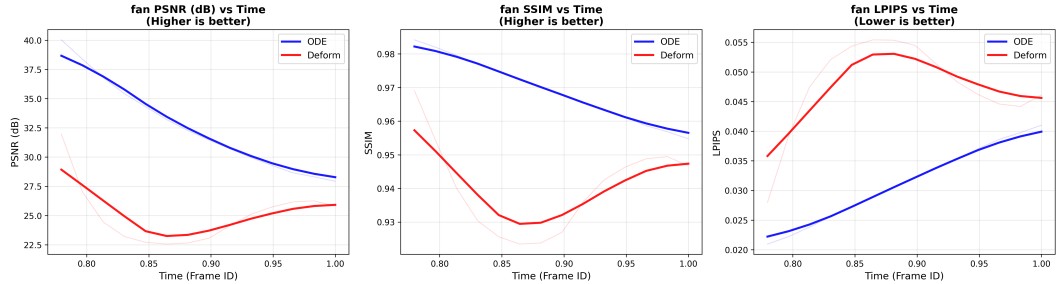

Figure 13: Metric performance on extrapolation over time compared with Deform-GS on scene fan from NVFi

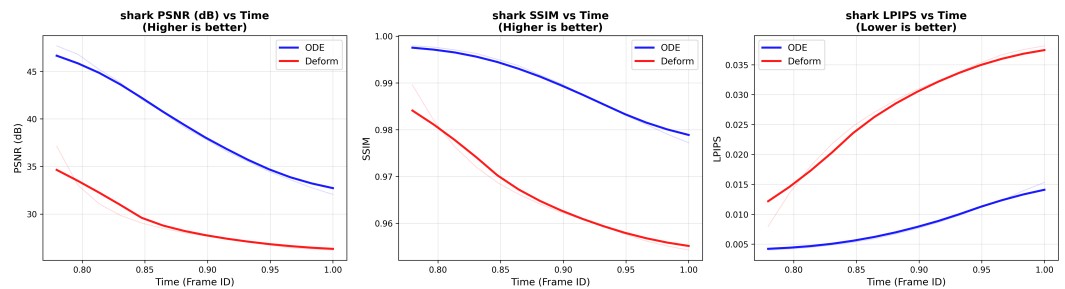

Figure 14: Metric performance on extrapolation over time compared with Deform-GS on scene shark from NVFi

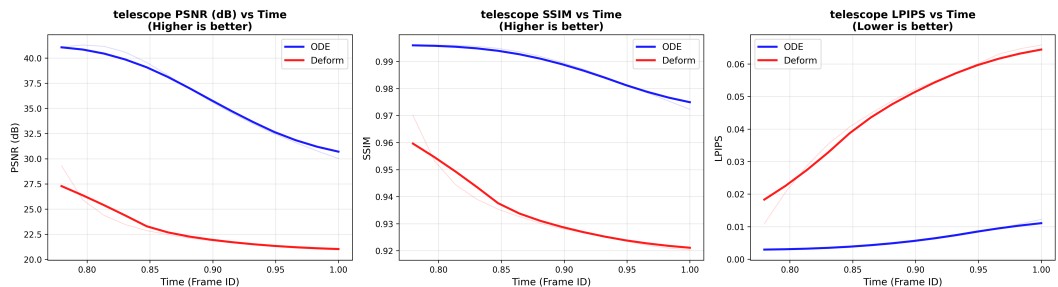

Figure 15: Metric performance on extrapolation over time compared with Deform-GS on scene telescope from NVFi

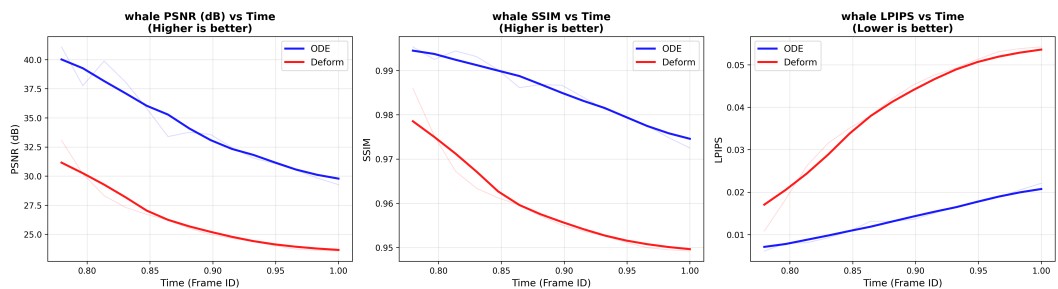

Figure 16: Metric performance on extrapolation over time compared with Deform-GS on scene whale from NVFi

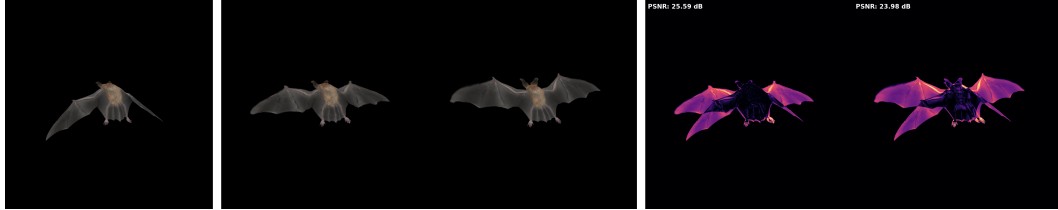

Figure 17: Qualitative comparison of last frame from scene bat of NVFi by only using an autoregressive transformer without the ODE component. From left to right is the GT image, the transformer's rendered image, the baseline (Yang et al., 2024)'s result, the difference between the GT and transformer's rendered image, and finally the difference between the GT and the baseline's rendered image.

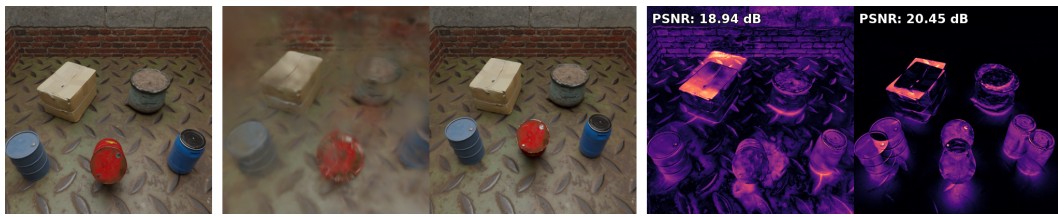

Figure 18: Qualitative comparison of last frame from scene factory of NVFi by only using an autoregressive transformer without the ODE component. From left to right is the GT image, the transformer's rendered image, the baseline (Yang et al., 2024)'s result, the difference between the GT and transformer's rendered image, and finally the difference between the GT and the baseline's rendered image.

Table 11: Quantitative extrapolation results on the **bouncingballs** scene of D-NeRF dataset. Comparison between Deformable-GS and Ours across different data splits. The best metric is highlighted in red. For Ours, we set the input time span to be a shorten span of 0.1 instead of the default 0.6.

| Method | Split 65% | | | Split 70% | | | Split 75% | | | Split 80% | | |
|---|---|---|---|---|---|---|---|---|---|---|---|---|
| | PSNR(↑) | SSIM(↑) | LPIPS(↓) | PSNR(↑) | SSIM(↑) | LPIPS(↓) | PSNR(↑) | SSIM(↑) | LPIPS(↓) | PSNR(↑) | SSIM(↑) | LPIPS(↓) |
| Deformable-GS | 24.11 | .9614 | .0533 | 23.30 | .9600 | .0550 | 24.08 | .9642 | .0478 | **29.49** | **.9804** | **.0237** |
| **Ours** | **26.55** | **.9690** | **.0130** | **26.27** | **.9679** | **.0158** | **26.84** | **.9715** | **.0146** | 28.02 | .9761 | .0357 |
| Method | Split 85% | | | Split 90% | | | Split 95% | | | Average | | |
| | PSNR(↑) | SSIM(↑) | LPIPS(↓) | PSNR(↑) | SSIM(↑) | LPIPS(↓) | PSNR(↑) | SSIM(↑) | LPIPS(↓) | PSNR(↑) | SSIM(↑) | LPIPS(↓) |
| Deformable-GS | 31.44 | .9842 | .0197 | 34.50 | .9886 | .0148 | 34.31 | .9889 | .0131 | 28.89 | .9772 | .0325 |
| **Ours** | **32.97** | **.9860** | **.0054** | **36.23** | **.9913** | **.0039** | **39.18** | **.9944** | **.0035** | **31.00** | **.9796** | **.0138** |

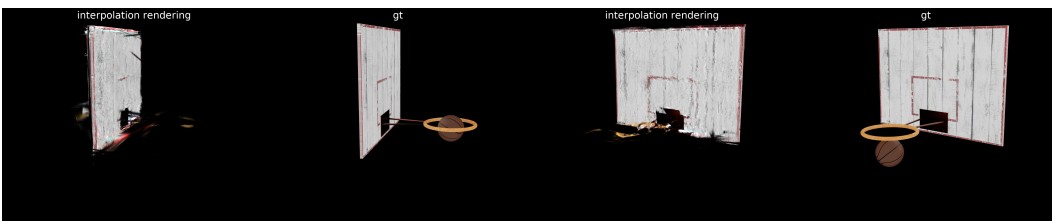

Figure 19: Qualitative comparison of the fallingball scene in NVFi on the interpolation model and the ground truth. As shown, the interpolation model fails to capture meaningful dynamics of the scene while missing the fallingball object.

