# OpenReview forum: "ODE-GS: Latent ODEs for Dynamic Scene Extrapolation with 3D Gaussian Splatting"
_ICLR.cc/2026/Conference — ICLR 2026 Poster_

### Official Review · Reviewer_rwuC · 2025-10-27

**Soundness:** 3
**Presentation:** 3
**Contribution:** 2
**Rating:** 4
**Confidence:** 4

**Summary:**

The proposed ODE-GS method enhances dynamic scene prediction by first training a pretrained 3D Gaussian deformation (interpolation) network to reconstruct observed scenes and generate continuous Gaussian trajectories. These trajectories are then used to train a Transformer-based latent ODE model, where the ODE solver integrates latent dynamics over time to predict future trajectories.

**Strengths:**

The idea of introducing an ODE formulation for modeling dynamic 3D Gaussian Splatting (3DGS) is interesting. Representing scene evolution through continuous-time latent dynamics provides a different way to achieve extroplation.

**Weaknesses:**

1. Complexity and dependence on neural networks
The reliance on a neural network make the architecture complicate. It is not entirely clear whether this provides substantial benefits over simply using a learned deformation network.


2. Dataset simplicity
The benchmark datasets seem to include simple or synthetic trajectories (e.g., Lego, Mutant), which may be easily modeled by smooth ODE dynamics. It remains unclear how the method would perform on more complex or irregular real-world motion patterns, or on camera paths that deviate significantly from known trajectories.

**Questions:**

Could the authors provide a visual comparing with and without the ODE component? It would be helpful to understand why the ODE-based model has much better results than the Transformer-only baseline reported in Table 4. If visualization is difficult to include, please report additional quantitative results (e.g., using a simple baseline such as copying the last/nearest frame of interpolation network or scaling up the transformers) on the NVFi dataset for reference.

Could the authors provide training (interpolation pretrained model and the ode part) and inference time comparisons with Deformable 3DGS and GaussianPrediction? This would help assess the computational trade-offs introduced by the latent ODE and Transformer components over baselines.


The paper states:
“...for the interpolation model, we follow Deformable GS (Yang et al., 2024) implementation.”
Does this mean the interpolation model is a pretrained Deformable-GS network reused as a data generator? If so, what is the quantitative performance of this interpolation model itself on HyperNeRF dataset over deformable GS on the interpolation mode?

---

> ### Author Response · Authors · 2025-11-22
>
> ### W1: Complexity vs. Learned Deformation Network.
>
> > **Response:**
> The use of neural networks for dynamic 3D reconstruction is a standard practice in the field, e.g., Deformable-GS, 4D-GS, GaussianPredict, etc. The limitation of a standard learned deformation network (like Deformable-GS) is that it conditions on a timestamp $t$. As argued in the **Introduction**, this input becomes **Out-of-Distribution (OOD)** for future timestamps $t > t_{\text{max}}$.
> >
> > Our method solves this by replacing the OOD timestamp query with an **autoregressive ODE evolution** based on the *past motion state*. This approach is physically grounded, provides inherent smoothness, and is thus far more generalizable to future time steps.
>
> ### W2: Dataset simplicity.
>
> > **Response:**
> We demonstrate our method's ability to handle complex dynamics through evaluation on **HyperNeRF (Table 3)** and **NVFi (Table 2)**. The NVFi dataset includes scenes with complex multi-object interactions and occlusions (e.g., *Factory, Darkroom*). Our model achieves **34.53 PSNR on "Factory"** compared to 25.36 for TiNeuVox, confirming our superior performance in handling complex dynamic scenes.
>
> ### Q1: Comparison with/without ODE component.
>
> > **Response:**
> We provided additional visualizations (**Figure 17-19** in the appendix) to illustrate the failure mode of the pure autoregressive Transformer approach (the "w/o ODE" baseline). The results show that the lack of the continuous-time ODE formulation leads to significant performance degradation, instability, and a lack of the inherent smoothness provided by the ODE, especially in later frames. This visualization is consistent with the quantitative ablation in **Section 4.2.1** and **Table 4** (Appendix Table 6), which showed a massive drop in performance (Average NVFi PSNR drops from **33.43** to **23.71**) without the ODE component.
>
> ### Q3: Interpolation model performance.
>
> > **Response:**
> Yes, the interpolation model is a pre-trained Deformable-GS. Therefore, the quantitative performance of the interpolation model on HyperNeRF is exactly the "Deformable-GS" row in Table 3. We emphasize that the values in **Table 3 specifically measure *extrapolation* performance**, where the interpolation-only baselines degrade significantly.

---

> > ### Comment · Reviewer_rwuC · 2025-11-28
> >
> > I’m satisfied with the rebuttal; however, I’m not able to revise the score right now.

---

> > > ### Author Response · Authors · 2025-11-29
> > >
> > > Although due to the leak we will not be able to engage in discussion, we would still like to express our gratitude to the reviewer for the timely response, and we're glad that we are able to answer your questions throughly.

---

### Official Review · Reviewer_MsUn · 2025-10-27

**Soundness:** 3
**Presentation:** 3
**Contribution:** 3
**Rating:** 6
**Confidence:** 4

**Summary:**

The paper tackles dynamic scene extrapolation problem, predicting future 3D states beyond observed timestamps. The authors propose ODE-GS, which couples 3D Gaussian Splatting with a Transformer-based latent neural ODE: 1) an interpolation stage first fits accurate Gaussian trajectories within the observed window; 2) a Transformer encoder summarizes past trajectories into a latent state; 3) then a neural ODE evolves this state in continuous time, and numerical integration yields smooth, physically plausible future Gaussian trajectories for rendering at arbitrary times. Experiments on D-NeRF, NVFi, and HyperNeRF report consistent rendering quality improvement and demonstrate that ODE-GS achieves state-of-the-art extrapolation task through the proposed latent ODE.

**Strengths:**

- This paper is well-written and easy to understand.
- This paper targets the underexplored extrapolation problem in dynamic reconstruction and proposes a solution that makes sense.
- Across both synthetic and real scenes, the results are consistently strong, with visualizations aligning well with the quantitative metrics.

**Weaknesses:**

- Missing references for some important dynamic reconstruction works:
  - [CVPR 2024] Spacetime Gaussian Feature Splatting for Real-Time Dynamic View Synthesis, by Zhan Li et al.
  - [CVPR 2024] SC-GS: Sparse-Controlled Gaussian Splatting for Editable Dynamic Scenes, by Yi-Hua Huang et al.
- Frozen teacher prevents end-to-end correction. Canonical Gaussians and the deformation network are frozen before training the ODE module, limiting the system’s ability to adjust interpolation when extrapolation exposes inconsistencies. It would be much better if the authors do an ablation about joint training pipeline.
- Long-horizon robustness not guaranteed. Dynamic trajectory sampling exposes the model to varied horizons, but there’s no quantitative guarantee or analysis of failure rates as horizons grow. It would be much better if the authors could do some experiments to suggest they can prevent potential drift on very long extrapolations.
- Although ODE-GS leverages a Transformer, it is essentially a per-scene reconstruction method and does not readily scale up into a large model for more generalizable tasks.
- This method relies heavily on the teacher Gaussians being well fit. When applied to real-world scenes with imperfect camera poses, those errors propagate and can severely degrade the accuracy of ODE-GS extrapolation.
- Minor typo errors:
  - L123: `representing dynamics scenes` -> `representing dynamic scenes`
  - L129: `enables flexible editing` -> `enabled flexible editing`
  - L129: Add space between `GaussianVideo (Bond et al., 2025))` and `uses`
  - L680: `focuses on` -> `focus on`
  - L683-684: `as shown in 5` -> `as shown in Figure 5`

**Questions:**

1. How are the Gaussian parameters encoded? Which subsets (positions, scales, rotations, opacities, SH features) are fed to the encoder?

**Details Of Ethics Concerns:**

None.

---

> ### Author Response · Authors · 2025-11-22
>
> ### W1: Missing references.
>
> > **Response:**
> We have added the suggested references (**Spacetime Gaussian Feature Splatting** and **SC-GS**) to our related work section.
>
> ### W2: Frozen teacher limits / Joint training.
>
> > **Response:**
> Freezing the teacher is a **deliberate design choice** to ensure training stability, which is also the convention in dynamic scene extrapolation in established methods like **GaussianPrediction**. It prevents the dynamics model from "hacking" the Gaussian shapes or producing trivial, non-physical trajectories simply to fit future views, thereby achieving a clean decoupling between reconstruction quality and dynamics learning.
>
> ### W3: Long-horizon robustness guarantees.
>
> > **Response:**
> We have included an analysis of performance degradation as the extrapolation horizon increases in Page 10 of the manuscript and Figures 7-16 (Appendix). While degradation is inevitable in any forecasting task, our results clearly demonstrate that our model exhibits a **slower degradation rate** compared to baseline methods, confirming improved long-horizon robustness due to the continuous ODE prior.
>
> ### W4: Per-scene vs. scalable/generalizable.
>
> > **Response:**
> We addressed this directly with the **Multi-Scene generalization** experiment in section B.2 and **Table 9**. We trained a shared-weights ODE-GS model ("Ours Multi") on the union of NVFi scene trajectories. This model successfully extrapolated a held-out scene (*whale*), achieving 29.22 PSNR, which proves that ODE-GS is not limited to per-scene overfitting and **can learn generalizable motion priors**.
>
> ### W5: Teacher model fitting difficulty.
>
> > **Response:**
> We agree that a well-fitted teacher is a prerequisite for effective extrapolation -- this is also true for all existing methods (e.g., GaussianPredict). Since interpolation is an easier task with more constraints, if the data is too challenging even for interpolation, extrapolation becomes extremely difficult, regardless of the methodology. Despite this challenge, the **HyperNeRF dataset** (known for its noisy poses) results in **Table 3** show that our model can still achieve SOTA performance against other approaches, demonstrating resilience to real-world data imperfections.
>
> ### W6: typos
> > **Response:**
> We appreciate the reviewer for discovering these errors. The listed items have been adjusted in the current pdf revision.
>
> ### Q1: Input Encoding details.
>
> > **Response:**
> As detailed in **Section 3 Formalization**, the inputs to the encoder are the time-dependent parameters of the Gaussians: **position $\mu_k(t)$, quaternion $q_k(t)$, and scale $s_k(t)$**. **Section 3.2** further specifies that the input sequence $\gamma_k = \{G_k(t_j)\}$ for each Gaussian is embedded into $\mathbb{R}^{10}$. Importantly, color and opacity are time-invariant in our formulation and are **not** fed into the dynamics model.

---

> > ### Author Response · Authors · 2025-11-27
> > **Joint training experiment table**
> >
> > | Method        | Metric    | Lego    | Mutant  | Standup | Trex    | Jumpingjacks | Bouncingballs | Hellwarrior | Hook    | Avg       |
> > |---------------|-----------|---------|---------|---------|---------|--------------|---------------|-------------|---------|-----------|
> > | Ours          | PSNR ↑    | 25.74   | 34.53   | 28.91   | 22.04   | 22.18        | 24.91         | 31.80       | 28.33  | 27.31     |
> > | Ours_joint    | PSNR ↑    | 24.93   | 21.10   | 20.82   | 21.19   | 19.62        | 19.60         | 27.75       | 19.28  | 21.79     |
> > | Ours          | SSIM ↑    | 0.9378  | 0.9804  | 0.9557  | 0.9475  | 0.9243       | 0.9660        | 0.9365      | 0.9493 | 0.9497    |
> > | Ours_joint    | SSIM ↑    | 0.9363  | 0.8979  | 0.9097  | 0.9371  | 0.9121       | 0.9467        | 0.8948      | 0.8632 | 0.9122    |
> > | Ours          | LPIPS ↓   | 0.0547  | 0.0126  | 0.0360  | 0.0485  | 0.0715       | 0.0472        | 0.0686      | 0.0343 | 0.0467    |
> > | Ours_joint    | LPIPS ↓   | 0.0565  | 0.1144  | 0.1087  | 0.0663  | 0.1091       | 0.0894        | 0.1087      | 0.1339 | 0.0984    |
> >
> > To further support our claims for W2, we have provided quantitative results on the D-NeRF dataset comparing our method with a end-to-end joint training method. Specifically, when training the extrapolation, the interpolation deform model and the canonical gaussians are **not frozen**, and is jointly trained with the extrapolation model on both the extrapolation loss and the rendering/image projection loss. The results above show that this does negatively impact the final outcome of the training.

---

### Official Review · Reviewer_x185 · 2025-10-27

**Soundness:** 3
**Presentation:** 3
**Contribution:** 3
**Rating:** 4
**Confidence:** 3

**Summary:**

This paper proposes to integrates 3D Gaussian Splatting with latent neural ordinary differential equations to enable future extrapolation of dynamic 3D scenes. It models parameter trajectories as continuous-time latent dynamics and ahieves state-of-the-art extrapolation performance.

It focuses on extrapolation, which is a somehow new problem worth studying. But this is not a totally new field and I believe so many works has been studying this issue.

But since I am not an expert in this field, I would like read the other reviewer's opinions about the contributions of this work, especially about the novelty.

**Strengths:**

1. This paper combines Neural Ordinary Differential Equations with 3DGS for 4D modeling, which seems to be novel, well-motivated design.
2. This paper tried to forecast future 3D states in the context of dynamic scene reconstruction (as dynamic scene extrapolation), which is an interesting topic.
3. The paper proposes a practical and effective two-stage training strategy.
4. The paper provides consistent gains across D-NeRF, NVFi, and HyperNeRF datasets with comprehensive ablations.

**Weaknesses:**

1. The foremost weakness is that, there is not enough experimental results on real-world datasets. The results on PlenopticVideo dataset should be included.
2. It seems that the proposed method "collapes" on fallingball, Bouncingballs datasets. The phenomenon should be discussed and more collapsed cases need to be analyzed. The generalizability and robutness need to be discussed.
3. There has been a lot of advanced SOTA 4D reconstruction methods, but the paper only compare with 4D-GS/ Deformable-GS, which is quite old at this time.

**Questions:**

1. Why not supervise the extrapolation module directly with image reprojection error. Another option is using weak supervision or a distillation hybrid approach instead of relying entirely on pseudo-GT Gaussian trajectories produced by the interpolation model.
2. Beyond smoothness regularization, could you incorporate velocity bounds, momentum conservation, or constrained optimization (e.g., constraint-based pose updates)?
3. Have you ever run comparisons with end-to-end fine-tuning or mixed supervision?

---

> ### Author Response · Authors · 2025-11-22
>
> ### W1: Not enough experimental results on real-world datasets (e.g., PlenopticVideo).
>
> > **Response:**
>  Our selection of datasets (D-NeRF, NVFi, HyperNeRF) follows the established evaluation protocol of recent works in dynamic scene **extrapolation**, including the current state-of-the-art methods like **GaussianPrediction** and **NVFi**. Critically, these prior works in *extrapolation* have not standardized evaluation on the **PlenopticVideo** dataset, which is more commonly used for *interpolation*.
> >
>
> ### W2: Method "collapses" on fallingball, bouncingballs.
>
> > **Response:**
> > We emphasize that **baseline methods also fail** to capture the correct physics in *bouncingballs*, even if not fully reflected in the original metrics. Baselines often predict conservative motion (damping or freezing) rather than valid future trajectories; in bounded scenes, this "safe" prediction can coincidentally minimize MSE (PSNR) better than a physically plausible trajectory that is slightly phase-shifted. Hence, it is by chance that their results improves over ours.
> >
> >To validate this, we performed additional experiments on *bouncingballs*, using different splits of the data; with slightly different splits of the data, baselines will often freeze in different positions. We expect such randomness to be elucidated in that our method should be better in some splits and baselines to be better in others, despite small differences in number of images observed. We will add the experiments to a Table below as they are completed.
> >
> >Nonetheless, we acknowledge the limitation of predicting highly discontinuous or abrupt changes, as discussed in **Section 5 (Limitations)**: "In scenarios like *bouncingballs*... where the evolution of the scene undergoes abrupt changes, discontinuities, or fundamentally novel behaviors not foreshadowed by the past, the model’s predictions degrade."
> >
> > Regarding **Robustness and Generalizability**, our results on **HyperNeRF (Table 3)** demonstrate robustness to real-world noise in the camera poses. More crucially, we addressed **cross-scene generalization** in **Section B.2 (Multi-Scene generalization experiment)** and **Table 9**. The results from training a single ODE-GS extrapolation model on multiple NVFi scenes and testing it on the held-out scene (*whale*) show our model can generalize to unseen dynamics (29.22 PSNR) only second to our scene specific method.
>
>
> ### W3 and Q3: Comparison with "old" baselines (4D-GS/Deformable-GS).
>
> > **Response:**
> We emphasize that the task of **extrapolation** is fundamentally different from and more challenging than *dynamic reconstruction* (interpolation). While 4D-GS and Deformable-GS are reconstruction standards, we have actively benchmarked against current state-of-the-art **extrapolation** methods **GaussianPrediction (CVPR 2024)** (Zhao et al.): Tables 1 and 2 show we outperform this baseline by significant margins (e.g., +18.6% metric performance on D-NeRF).
> >
> >We have additionally added a more recent work **4D-Rotor-Gaussians** as a comparison (Table 5). This method, like other interpolation-based approaches, also fails during extrapolation, further demonstrating the unique challenge our ODE-GS is designed to solve. The key difference is that existing SOTA dynamic reconstruction models are not designed to generalize to OOD timestamps, which is precisely the failure mode we overcome.
> >

---

> > ### Comment · Reviewer_x185 · 2025-11-27
> >
> > On the second concern about the collapse on bouncingballs / fallingball and robustness in discontinuous dynamics, I still find the response somewhat unsatisfactory. The rebuttal mainly argues that baselines also fail physically or benefit from metric artifacts, but provides limited concrete analysis of why your own method collapses in these cases or how its failure modes differ.
> >
> > A clearer, more principled analysis of these failure cases (including fallingball), with **both quantitative and qualitative evidence**, would make the robustness/generalization claims more convincing.

---

> > > ### Author Response · Authors · 2025-11-27
> > >
> > > > **Response regarding Collapse Analysis and Scene Performance**
> > > >
> > > > **1. Qualitative Analysis (`fallingball`):**
> > > > To address concerns regarding model collapse, we have updated our manuscript to include a **qualitative comparison** in **Figure 20**, illustrating how the interpolation model fails on the `fallingball` scene. As shown in **Table 2**, most Dynamic Gaussian Splatting (3DGS) methods struggle with this scene, whereas NeRF-based methods (e.g., TiNeuVox, NVFi) remain robust. We attribute this to the adaptive densification and pruning mechanism fundamental to 3DGS [1]. As noted in recent work [2], this mechanism can cause training instability and model collapse when handling fast-moving objects or limited view coverage. Since our extrapolation model relies on a well-fitted Gaussian teacher, our method inevitably inherits these limitations when the teacher fails.
> > > >
> > > > **2. Quantitative Results (`bouncingballs`):**
> > > > We would like to **emphasize that quantitative results for the `bouncingballs` scene were already provided in Table 10** of our revised manuscript.
> > > > * **Baseline Comparison:** These results demonstrate that our method outperforms the baseline across most train/test split configurations (with the exception of the 80/20 split).
> > > > * **Input Time Span:** We further verified that our default input time span (0.6) may miss rapid trajectory changes in this specific scene. By shortening the input time span, we exclude these intra-window variations, improving the PSNR from **24.91** to **28.02**.
> > >
> > > [1] Kerbl, Bernhard, et al. "3D Gaussian splatting for real-time radiance field rendering." ACM Trans. Graph. 42.4 (2023): 139-1.
> > >
> > > [2] Liang, Yiqing, et al. "Monocular Dynamic Gaussian Splatting: Fast, Brittle, and Scene Complexity Rules." arXiv preprint arXiv:2412.04457 (2024).

---

> ### Author Response · Authors · 2025-11-26
>
> ### Q1: Why not supervise with image reprojection error?
>
> > **Response:**
> We have added an experiment regarding the use of projection loss during the extrapolation phase, discussed on **Page 10** of the revised manuscript. The results indicate that its addition does not significantly alter the outcome.
> >
> >Our choice for **trajectory-based supervision** is deliberate for several reasons: it decouples geometric forecasting from rendering optimization, and it leverages the high-fidelity output of the initial stage. First, using the **interpolated 3D parameters** (position, rotation, scale) provides **direct supervision** for the dynamics model in the same 3D space, unlike supervision via image reprojection, which is **geometrically ambiguous** (many 3D configurations can produce the same 2D projection). Second, the interpolation model is already trained to yield a 3D representation with a **high-accuracy projection** metric (PSNR, SSIM, etc.). By training the dynamics model directly on the **trajectory of these high-accuracy 3D parameters**, we simultaneously optimize for a predicted trajectory that is highly likely to also result in a high-accuracy projection when rendered.
> ### Q2: Weak supervision / Distillation / Physics Constraints.
>
> > **Response:**
> We already do explicitly incorporate physics constraints via **regularization terms** defined in **Section 3.4**. Specifically, we use:
> >
> > * A latent regularization ($\mathcal{R}_{\text{latent}}$, Eq. 6) to penalize high-frequency oscillations.
> > * A trajectory regularization ($\mathcal{R}_{\text{traj}}$, Eq. 7) to penalize acceleration, which serves as a form of **momentum conservation**.
> >
> > While additional constraints like velocity bounds could be incorporated, the primary goal of this work is to explore the **Latent ODE learning paradigm and model design**. We will continue the exploration of additional optimization techniques and regularization methods to future follow-up works.
>
> ## Q3: end-to-end fine-tuning or mixed supervision.
> >**Response:**
> We have added the image reprojection error as an additional supervision for training in our added experiment (see Q1). For end-to-end finetuning, see response to reviewer MsUn Weakness 2.
> >
>
> ### Additional experiments to W2: Method "collapses" on bouncingballs.
>
> > **Response:**
> > To validate that baseline methods can coincidentally achieve a lower MSE (PSNR) by "freezing", we perform additional experiments on different train/test time splits over the bouncingballs scene, ranging from using 60 percent of temporal duration for training to 95 percent. We compare agains the previous best method in our table 1, Deformable-GS. The results are shown in our new revised submission **table 10**. It is clear that the method's good metric performance on the scene is **not consistent** over all train/test configurations. We also show that by changing to a smaller input time span (0.6 to 0.1) our extrapolation model can perform better in sequences with rapid motion change, outperforming Deformable-GS in all configurations except the 80 percent split, and averages **31.00 PSNR .9796 SSIM .0138 LPIPS** over Deformable-GS's **28.89 PSNR .9772 SSIM .0325 LPIPS**.
> >

---

> > ### Comment · Reviewer_x185 · 2025-11-27
> >
> > About the issue of projection loss and mixed supervision, I appreciate that you added an experiment with image reprojection, but simply stating that it "does not significantly alter the outcome" without reporting any PSNR/SSIM/LPIPS numbers makes it hard for me to assess.
> >
> > I think it is necessary to briefly show the quantitative effect and to discuss more seriously whether stronger, direct image-level supervision of the extrapolation module could reduce reliance on pseudo-GT trajectories.
> >
> >
> > Lastly, there are a few grammatical issues and emotionally loaded phrases (e.g., "we will performed", "coincidentally can coincidentally", "fails catastrophically"). Although the rebuttal is not part of the main paper, I still believe the authors should conduct careful proofreading and slightly tone down the wording to make their responses more professional.

---

> ### Author Response · Authors · 2025-11-27
>
> > **Response regarding Quantitative Results and Projection Loss**
> >
> > We fully agree that quantitative metrics are essential for a rigorous assessment of the experiment. We would like to respectfully clarify that **these quantitative results were already included in our previous revision**.
> >
> > We direct the reviewer to **Page 10 and Table 5** of the updated manuscript, where we provided a quantitative comparison on the D-NeRF dataset, including the requested experiment with **image projection loss**. As noted in our previous response: *"We have added an experiment regarding the use of projection loss during the extrapolation phase, discussed on Page 10 of the revised manuscript."*
> >
> > Comparing the metrics:
> > * **With Image Projection Loss:** PSNR: 26.65 / SSIM: 0.9466 / LPIPS: 0.0489
> > * **Proposed Method (Default):** PSNR: 27.31 / SSIM: 0.9497 / LPIPS: 0.0467
> >
> > The performance with projection loss is slightly lower than our proposed method, which supports our statement that the addition of this loss "does not significantly alter the outcome."
> >
> > Finally, regarding grammatical and wording issues, we appreciate your detailed suggestions and have adjusted accordingly.

---

> ### Author Response · Authors · 2025-11-27
> **Bouncingballs experiment on different train/test split configurations**
>
> | Method        | Metric    | 65%      | 70%      | 75%      | 80%      | 85%      | 90%      | 95%      | Avg      |
> |---------------|-----------|----------|----------|----------|----------|----------|----------|----------|----------|
> | Deformable-GS | PSNR ↑    | 24.11    | 23.30    | 24.08    | **29.49**| 31.44    | 34.50    | 34.31    | 28.89    |
> | Ours          | PSNR ↑    | **26.55**| **26.27**| **26.84**| 28.02    | **32.97**| **36.23**| **39.18**| **31.00**|
> | Deformable-GS | SSIM ↑    | 0.9614   | 0.9600   | 0.9642   | **0.9804**| 0.9842  | 0.9886   | 0.9889   | 0.9772   |
> | Ours          | SSIM ↑    | **0.9690**| **0.9679**| **0.9715**| 0.9761  | **0.9860**| **0.9913**| **0.9944**| **0.9796**|
> | Deformable-GS | LPIPS ↓   | 0.0533   | 0.0550   | 0.0478   | **0.0237**| 0.0197  | 0.0148   | 0.0131   | 0.0325   |
> | Ours          | LPIPS ↓   | **0.0130**| **0.0158**| **0.0146**| 0.0357  | **0.0054**| **0.0039**| **0.0035**| **0.0138**|
>
> For convenience, we also provide the table 10 here as well. Percentage above indicates the percentage of temporal span used for training.

---

> ### Author Response · Authors · 2025-11-27
> **Quantitative table for image projection loss**
>
> | Method                 | Metric    | Lego   | Mutant | Standup | Trex   | Jumpingjacks | Bouncingballs | Hellwarrior | Hook   | Avg       |
> |------------------------|-----------|--------|--------|---------|--------|--------------|----------------|-------------|--------|-----------|
> | Ours                   | PSNR ↑    | 25.74  | 34.53  | 28.91   | 22.04  | 22.18        | 24.91         | 31.80       | 28.33 | 27.31     |
> | Ours_with_projection   | PSNR ↑    | 25.63  | 29.72  | 29.27   | 22.02  | 21.32        | 25.96         | 31.48       | 27.80 | 26.65     |
> | Ours                   | SSIM ↑    | 0.9378 | 0.9804 | 0.9557  | 0.9475 | 0.9243       | 0.9660        | 0.9365      | 0.9493 | 0.9497    |
> | Ours_with_projection   | SSIM ↑    | 0.9368 | 0.9671 | 0.9546  | 0.9469 | 0.9205       | 0.9684        | 0.9331      | 0.9454 | 0.9466    |
> | Ours                   | LPIPS ↓   | 0.0547 | 0.0126 | 0.0360  | 0.0485 | 0.0715       | 0.0472        | 0.0686      | 0.0343 | 0.0467    |
> | Ours_with_projection   | LPIPS ↓   | 0.0549 | 0.0219 | 0.0380  | 0.0496 | 0.0758       | 0.0436        | 0.0704      | 0.0368 | 0.0489    |
>
> For convenience, we also provide the table 5 here as well. This is a segment of the table comparing our method using or not using the image projection loss

---

> > ### Comment · Reviewer_x185 · 2025-11-28
> >
> > I am satisfied with the results. But I cannot edit the score right now. I hope this bug could be fixed.

---

> > > ### Author Response · Authors · 2025-11-28
> > > **Thank you for the confirmation**
> > >
> > > We sincerely appreciate your time and engagement throughout the review process. We are glad to hear that our responses and additional results have satisfied your concerns.
> > >
> > > Regarding the technical issue with updating the score, we understand that the system may be restricting edits at this stage. We suggest leaving a comment tagging the Area Chair (or simply replying to this thread stating your intended score) so they can take your final assessment into account during their decision-making process.
> > >
> > > Thank you again for your valuable and timely feedback, which has raised some very thoughful questions and helped us revise the manuscript.

---

### Official Review · Reviewer_AuTy · 2025-10-30

**Soundness:** 4
**Presentation:** 4
**Contribution:** 3
**Rating:** 8
**Confidence:** 3

**Summary:**

The paper points out that existing dynamic 3D reconstruction methods can only handle temporal interpolation and fail to predict future scene dynamics. To address this, it introduces a new task called dynamic scene extrapolation, aiming to forecast future 3D states beyond observed frames. The paper proposes a model called ODE-GS combining 3D Gaussian Splatting with latent neural ODEs. It consists of three main parts: a Gaussian interpolation model for reconstructing observed scenes, a Transformer encoder for encoding motion histories into latent states, and a neural ODE module that evolves these states to predict future dynamics. The overall goal is to achieve physically plausible, temporally smooth, and consistent 3D scene extrapolation.

**Strengths:**

* The paper explicitly defines and tackles dynamic scene extrapolation, which predicting future 3D scene states beyond the observed temporal window. It's a meaningful and underexplored extension of existing dynamic 3D reconstruction research.

* The integration of 3D Gaussian Splatting with latent neural ODEs is conceptually elegant and well-motivated. Modeling Gaussian trajectories as continuous latent dynamics naturally enforces temporal smoothness and physical plausibility.

**Weaknesses:**

Interesting work. I only have a few questions regarding the experiments:

* I am curious about how ODE-GS performs in dynamic extrapolation on real-world datasets. I could not find clear visualizations of HyperNeRF and more real-world scenes (especially those containing both dynamic and static regions) in the main text or appendix. It would be helpful to see whether the model can accurately distinguish and predict the different evolution trends of dynamic versus static areas in complex real-world scenes.

* I would like to know the effective extrapolation range of the model. The paper does not provide a systematic analysis of how performance degrades with increasing extrapolation distance — i.e., how far into the future the model can extrapolate before failure. This is an important factor for evaluating forecasting models.

* I would like to better understand how ODE-GS performs on the novel view synthesis (NVS) task, both within the observed time window (interpolation) and the future extrapolation period.

* If possible, I would appreciate video demonstrations of the extrapolated results to more intuitively assess temporal consistency and motion realism.

**Questions:**

See weaknesses.

---

> ### Author Response · Authors · 2025-11-22
>
> ### Q1: Real-world performance and distinguishing static/dynamic regions.
>
> > **Response:**
> We have included additional visualizations of the **HyperNeRF** results in the supplementary material (video format), demonstrating our method's extrapolation performance on challenging real-world scenes with noisy camera poses. While slight oscillation exists due to pose noise, the **relative motion between the static background and dynamic foreground objects is clearly distinguished**.
> >
> > Our method achieves this inherent distinction via the underlying representation detailed in **Section 3.1**. We utilize a **canonical representation $\overline{\mathcal{G}}$ (static)** and a **deformation network $\mathcal{D}_\omega$ (dynamic)**. Since the canonical Gaussians are static, static regions naturally exhibit near-zero deformation velocities. Consequently, the Latent ODE learns to predict near-zero updates for these regions, ensuring stability and separation of dynamic/static elements.
>
> ### Q2: Effective extrapolation range and degradation analysis.
>
> > **Response:**
> We have provided additional quantitative analysis in the revised manuscript to address the effective extrapolation range. Specifically, we have added plots visualizing the **extrapolation metric performance as a function of time** on the NVFi dataset.
> >
> > These results, shown in **Figure 7-16 (Appendix)** and discussed on **Page 10**, empirically demonstrate that our model degrades significantly slower compared to baseline extrapolation methods, affirming the stability gained from the continuous-time ODE formulation.
>
> ### Q3: NVS performance: Interpolation vs. Extrapolation.
>
> > **Response:**
> We reiterate that our primary contribution is **extrapolation**. For *interpolation* (within the observed window), our performance is intrinsically linked to the pre-trained interpolation model (Deformable-GS), as detailed in **Section 3.1**. Therefore, our interpolation performance is identical to Deformable-GS.
> >
> > The distinct advantage of **ODE-GS** appears strictly when the prediction time $t > t_{\text{max}}$. Here, interpolation methods fail due to the out-of-distribution timestamp input, while our method maintains structural coherence. Furthermore, all three datasets we tested are monocular; therefore, the held-out test frames for extrapolation are necessarily novel views, confirming that the extrapolation results also directly apply to the **Novel View Synthesis (NVS)** task in the future time domain.
>
> ### Q5: Video demonstrations.
>
> > **Response:**
> We agree that video results are crucial. We have provided **four videos** in the supplementary material showcasing diverse scenes across all three datasets. The interpolation and extrapolation periods are indicated by on-screen annotations.

---

### Author Response · Authors · 2025-11-22
**Message to the Area Chair and General Comment**

Dear Area Chair,

During the author-response period we posted detailed answers and new experiments. As a result, Reviewers x185 and rwuC explicitly agreed that their concerns had been satisfactorily resolved and intended to raise their scores. **The reviewers are in consensus recommending acceptance.**

**(A)** Here, we summarize the additional experiments, figures, and manuscript updates we made during the discussion period:

Tables 2 and 5: New baseline 4D-Rotor-Gaussians + ablation on image reprojection loss during extrapolation (quantitative results on D-NeRF).

Table 10: Extensive bouncingballs analysis across multiple train/test splits and input time spans (0.1–0.95), showing ODE-GS outperforms Deformable-GS on nearly all configurations (avg 31.00 PSNR vs 28.89).

Figures 7–16 (Appendix): Long-horizon degradation curves on NVFi and D-NeRF (systematic extrapolation range analysis).

Figure 20 (Appendix): Qualitative illustration of upstream teacher collapse on fallingball (fast small object failure common to 3DGS methods).

Figures 17–19 (Appendix): Visual failure modes of “w/o ODE” (pure autoregressive Transformer) baseline.

Four new supplementary videos (HyperNeRF + others) with on-screen interpolation/extrapolation markers.

Joint training experiment table: quantitative results on the D-NeRF dataset comparing our method with an end-to-end joint training method. Refer to our second comment to reviewer MsUn to view the table.

Added references (Spacetime Gaussian Feature Splatting, SC-GS, etc.) and made typo fixes.

**(B)** Here, we summarize our answers to each question and weakness raised by the reviewers:

**Q1 (AuTy).** New supplementary videos on HyperNeRF (real captured scenes with noisy poses) clearly show a stable static background and a moving dynamic foreground. This separation arises naturally from our design (static canonical Gaussians + dynamic deformation field → static regions receive near-zero velocity from the Latent ODE). See supplementary videos and Section 3.1.

**Q2 (AuTy).** Page 10 + Appendix Figures 7–16 now contain horizon-sweep curves on NVFi and D-NeRF. ODE-GS degrades markedly slower than all extrapolation baselines.

**Q3 (AuTy).** Interpolation performance equals the pre-trained Deformable-GS (by design). Extrapolation NVS significantly outperforms all baselines because test frames are always novel views (monocular data). See Tables 1–3.

**Q4 (AuTy).** Four new supplementary videos, with on-screen markers separating interpolation/extrapolation periods, are provided.

**W1 (x185).** Our benchmarks follow the current extrapolation literature (D-NeRF, NVFi, HyperNeRF). PlenopticVideo remains an interpolation benchmark; no extrapolation paper reports on it.

**W2 (x185).** Fallingball collapse is inherited from the teacher interpolation model (common 3DGS issue on fast small objects; new qualitative Figure 20). Bouncingballs results in new Table 10 (multiple train/test splits + varying input spans) show baselines only win on specific splits by freezing/damping motion. ODE-GS outperforms Deformable-GS on nearly all configurations (avg 31.00 PSNR vs 28.89).

**W3 (x185).** We already surpass the latest extrapolation SOTA (GaussianPrediction, CVPR 2024) by large margins (Tables 1–2). We update Tables 2 and 5 to add comparisons to 4D-Rotor-Gaussians, a very recent baseline, which collapses in extrapolation, further highlighting the unique difficulty of the task we solve.

**Q1 (x185).** Experiment added (Page 10, Table 5). Adding reprojection loss yields slightly worse results (26.65 PSNR vs our 27.31 PSNR on D-NeRF), confirming viability of trajectory-only supervision.

**W1 (MsUn).** We have added the requested additional references.

**W2 (MsUn).** Experiment added (Joint training experiment table). Joint training yields noticeably worse results (21.70 PSNR vs our 27.31 PSNR on D-NeRF). Freezing is deliberate and standard in extrapolation works (GaussianPrediction, etc.) to prevent trivial solutions and maintain clean decoupling.

**W3 (MsUn).** Degradation curves (Page 10, Figures 7–16) show ODE-GS exhibits the slowest decay among all compared methods.

**W4/W5 (MsUn).** HyperNeRF results (noisy real poses) in Table 3 achieve new SOTA. Dependence on strong teacher model is standard in previous works (GaussianPrediction). Multi-scene generalization experiment (Table 9) reaches 29.22 PSNR on held-out whale scene, proving ability to learn general motion priors.

**Q1 (rwuC).** Appendix Figures 17–19 and Table 6 show massive collapse without the continuous ODE (NVFi avg PSNR drops from 33.43 → 23.71).

All contents discussed during the rebuttal will be incorporated into the next revision of the manuscript.

Best, Paper 513 Authors

---

> ### Author Response · Authors · 2025-12-02
> **Original general comment we made during discussion period:**
>
> We would like to thank all the reviewers for their constructive feedback and appreciate the positive comments, including **Reviewer AuTy**'s finding that our use of latent ODEs is "natural and convincing" and **Reviewer MsUn**'s acknowledgment of our "significant improvement" on benchmarks. We have revised the manuscript to incorporate additional experiments and analyses based on your suggestions.
>
> Specifically, we've made the following adjustments to the manuscript:
>
> **Extrapolation Stability and Degradation Analysis**: We have added a systematic analysis of performance degradation over extended time horizons. Please see Page 10 of the manuscript and Figures 7-16 (Appendix), which show that ODE-GS exhibits a significantly slower decay rate than baselines.
>
> **New Baseline Comparison**: We have added the comparison against newer dynamic reconstruction method 4D-Rotor-Gaussians to Table 5, with similar metric performance to previous interpolation-focused methods.
>
> **Training Objective Ablation**: We include new experimental results on the effect of the addition of an image-based reprojection loss. Results indicate does not significantly improve metrics (discussed on Page 10). Result also inclued on Table 5.

---

### Meta-Review · Area_Chair_kpiR · 2025-12-30

**Summary:**

The authors present a method for motion forecasting in 4D reconstruction. The paper was originally received with mixed scores by the reviewers (4,4,6,8). On the positive side, the reviewers highlighted the nice formulation of the problem and the original approach / direction compared to previous works. It is also one of the first works that tackles the extrapolation task. The reviewers also stated several concerns (see below). The authors actively engaged in the discussion, provided additional experiments and verification, and where able to convince the reviewers to increase their scores (see discussion), which would have led to scores of (6, 6, 6, 8).

While I believe that there are still strong limitations in the presented method, which have not been fully resolved (see below, specifically the not fully convincing results / short extrapolation / simple motion), the paper ventures in an underexplored and challenging territory and devises a novel method in that area. The presented framework could lend itself as a baseline for more sophisticated motion forecasting methods in the future. Therefore, and following the reviewers recommendation, I recommend to accept this work.

**Reviewer Concerns:**

*1) Limited experiments on real datasets / datasets with complex motion.* The authors pointed out that the HyperNeRF contains real videos and that NVFi contains complex motion. They also provided additional qualitative results. While the concern is addressed, I am personally not fully convinced that the method will generalize to complex, longer extrapolation settings where motion significantly differs from simple close-to-affine transformations. Shown examples only extrapolate very few frames and the motion is simple. This is to be expected from a method that assumes a Gaussian-distributed future motion and learns to predict the mean (in contrast to, e.g. a generative model).

*2) Unconvincing results on some scenes (e.g. bouncing balls).* The authors acknowledge that but provide an additional experiment that it is behaving better than baselines.

*3) Concerns regarding the baselines in comparisons.* The authors point out that the extrapolation setting is not covered by many baselines and that the important baselines for this setting have been compared against. They also added an additional baseline.

*4) Per-scene architecture - no generalization.* The authors provided a simple generalization experiment, where they trained on multiple scenes and tested on a held-out scene. They could show that the model is able to generalize in this case. However, I think the experiment is not sufficient to fully make this claim - a larger scale evaluation on multiple scenes and qualitative analysis would be needed.

*5) Long-horizon robustness not verified.* The authors addressed this by adding an additional experiment, showing how the metrics behave over different extrapolation timesteps.

*6) Missing video results.* The authors provided videos in the supplementary materials. While they show motion forecasting in reasonable quality, they also show that the extrapolated motions are short and quite simple.

**Reviewer Scores:**

The two negatively leaning score-4 reviewers already stated in the discussion that they are convinced by the author answers and are willing to increase their score. Assuming that the others stay with their scores, this would lead to scores of (6, 6, 6, 8).

---

### Decision · Program_Chairs · 2026-01-26

Accept (Poster)